# A meta-interactive neural network for solving time-varying quadratic programming problems

Zhijun Zhang [1,2,3,4,5,6,7,8,9,10,12] ✉, Xiangliang Sun [1,12] ✉, Yiqi Liu[1] & Yamei Luo[11]

Many practical applications can be formulated as time-varying quadratic programming (TVQP) problems. Improving solution speed and accuracy can theoretically enhance efficiency. However, existing solvers such as the zeroing neural network (ZNN) and varying-parameter recurrent neural network (VPRNN) exhibit inherent limitations. Here, we propose a meta-interactive neural network (MINN). Unlike the independent neural structures in ZNN and VPRNN, the proposed MINN constructs a coupled topology for neurons, enabling information exchange within the network, and utilizing group dynamics to accelerate the convergence process. Notably, MINN relaxes the activation function constraints imposed by ZNN, allowing the use of non-monotonically increasing odd functions, thereby broadening the class of admissible activations. Lyapunov-based analysis confirms the enhanced convergence properties of MINN. Furthermore, numerical simulations demonstrate that MINN consistently outperforms ZNN and VPRNN in terms of convergence speed and robustness. Surprisingly, MINN also generalizes well to other time-varying problems, such as the Sylvester equation. Additionally, a detailed analysis of the coupling parameters reveals its critical role in system performance. Finally, applying MINN to robotic motion planning improves control accuracy from $10^{-6}$m to $10^{-7}$m.

In science and engineering, optimization problems (particularly quadratic programming problems) are widely encountered, such as robot motion planning[1], project selection in portfolio[2], signal processing[3], optimized resource allocation in distributed systems[4], and production scheduling in industrial manufacturing[5]. These seemingly disparate real-world problems share a common core challenge: rapidly finding the optimal solution that minimizes a quadratic objective function under dynamically changing constraints. A particularly challenging variant is time-varying quadratic programming (TVQP), where parameters evolve dynamically over time posing significant challenges for real-time computation and robustness[6–8]. Traditional numerical solvers often struggle to balance solution accuracy

[1]School of Automation Science and Engineering, South China University of Technology, Guangzhou, Guangdong, China. [2]Key Library of Autonomous Systems and Network Control, Ministry of Education, School of Automation Science and Engineering, South China University of Technology, Guangzhou, Guangdong, China. [3]Guangdong University Key Laboratory of Large-Model Embodied-Intelligent Humanoid Robot, Guangzhou, Guangdong, China. [4]Institute for Super Robotics (Huangpu), Guangzhou, Guangdong, China. [5]Nanchang University, Nanchang, Jiangxi, China. [6]College of Computer Science and Engineering, Jishou University, Jishou, Hunan, China. [7]Guangdong Artificial Intelligence and Digital Economy Laboratory (Pazhou Lab), Guangzhou, Guangdong, China. [8]School of Electronical Engineering, Shaanxi University of Technology, Hanzhong, Shaanxi, China. [9]School of Information Science and Engineering, Changsha Normal University, Changsha, Hunan, China. [10]Institute of Artificial Intelligence and Automation, Guangdong University of Petrochemical Technology, Maoming, Guangdong, China. [11]Department of Mechanical and Automation Engineering, The Chinese University of Hongkong, Hong Kong, China. [12]These authors contributed equally: Zhijun Zhang, Xiangliang Sun. ✉e-mail: auzjzhang@scut.edu.cn; sxl245578@163.com

with computational latency under time-varying constraints[9,10]. This has driven the development of neural network-based parallel computing strategies.

Among existing neural network-based solvers, the zeroing neural network (ZNN)[11] and varying-parameter recurrent neural network (VPRNN)[12] have emerged as dominant approaches. ZNN reformulates the TVQP problem into a system of differential equations, where each neuron corresponds to a variable in the optimization problem, and explicit kinetic equations derived from the error evolution are used to solve the problem. VPRNN enhances this framework by introducing time-varying growth parameters, thereby improving convergence speed and robustness. Currently, these models have been widely applied to robot motion planning[1,13-15] and the solution of time-varying problems[16-19].

However, both ZNN and VPRNN adopt decentralized topologies, where neurons operate independently without communication. While this design reduces computational complexity, it limits scalability and adaptability. Specifically, the absence of inter-neuronal communication restricts global information integration, causing convergence rates to remain uniform across all neurons. As a result, subsequent studies focused on improving the convergence speed of networks by adjusting convergence parameters to achieve super-exponential convergence[18,20,21]. For example, some approaches incorporated error-dependent adaptive mechanisms[18,20], while others investigated multiple time-varying growth parameters[21]. Nonetheless, these models remain confined by their independent neuron structures. Furthermore, both ZNN[11] and VPRNN[12] require monotonically increasing odd activation functions. However, this requirement precludes the application of other functions that do not meet the aforementioned conditions, such as the Swish activation function. Therefore, a more generalized model is required.

In contrast, biological neural systems achieve complex computation not through isolated processing units, but through dynamically coupled networks mediated by synaptic interactions[22-24]. Studies of the cerebral cortex shown that neurons integrate and modulate information via localized synaptic connectivity involving dendritic spines and axon terminals[25]. This coupled topology allows neurons to generate group dynamic behaviors such as collective oscillations and synchronized discharges through interacting synaptic inputs, enabling efficient information integration and error correction in cognitive processes[26]. For example, coordinated neural activity enables the seamless integration of multimodal inputs (e.g., vision and hearing) during motor control tasks.

Motivated by these insights, we propose a meta-interactive neural network (MINN) inspired by the coupled interaction mechanism of biological neurons and the Hopfield neural network (HNN)[27]. Unlike ZNN and VPRNN models, MINN incorporates dynamic interactions among neurons to facilitate distributed information exchange and cooperative computation. The coupled topology enables richer neuronal dynamics and a broader design space for network behaviors. Therefore, by leveraging the coupling relationship between neurons, MINN can exploit collective dynamics to accelerate convergence and enhance solution accuracy. A similar architecture has been successfully applied in system synchronization and nonlinear control, highlighting its potential for optimization problem solving. For example, similar coupled architectures have been used to compute the synchronization control of master-slave systems and applied to image encryption and decryption[28,29]. Additionally, similar frameworks have been utilized to address nonlinear system control[30,31]. In contrast to ZNN, where neurons merely act as communication nodes, MINN's coupled neurons facilitate distributed computation, making the network more scalable.

Comparison with related works: The MINN proposed in this paper differs from most existing models. Specifically, traditional research focuses on improving network performance by adjusting parameters

and activation function[12,18-21]. In contrast, MINN enhances computational efficiency by constructing communication topologies between neurons and leveraging the collective dynamics of neurons. This approach provides other perspective for existing research; based on MINN, better models are expected to be derived by adjusting network structures and coupling parameters. Furthermore, we note that the design of activation functions in refs. [11,12,16-18] is relatively simplistic, taking monotonically increasing odd functions as the sole criterion for selecting activation functions. Although this assumption simplifies mathematical proofs, its application scope is narrow. Through mathematical derivation and proof, this study expands the application scope of activation functions. In refs. [28-31], neural network models similar to MINN were used to solve control problems of nonlinear systems. However, it is evident that these neural networks are only applied to specific problems. The MINN proposed in this paper is a more general method. In addition to solving TVQP problems, it can be applied to other time-varying problems (e.g., the artificial intelligence, numerical computation, robotics, signal processing, control systems, and some specific examples can be seen from the Supplementary Information).

In summary, this work introduces a MINN framework for solving the TVQP problem. By integrating the advantages of neuronal coupling and dynamic modeling, the proposed MINN addresses the key limitations of conventional independently structured neural networks. Figure 1 shows the overall design framework of MINN. Furthermore, the main contributions of this paper are summarized as follows:

1. The contribution is that it imitates the information exchange behaviors between biological neurons, and construct a coupled topology for artificial neurons, which enables information exchange within the network, and utilizes group dynamics to accelerate the convergence process. It breaks the inherent pattern of traditional neural networks that can only improve network performance by fine-tuning parameters. Its significance lies in better matching the information exchange patterns of biological nervous systems and revolutionizing the weight update methods commonly used in existing neural networks.

2. A MINN is proposed for solving the TVQP problem. Unlike the independent neuron connection design in conventional ZNN[11] and VPRNN[12] models, MINN introduces an inter-neuronal communication topology that enables dynamic coupling among neurons, thereby enhancing convergence performance through collective behavior. Moreover, the influence of coupling parameters on system performance is systematically investigated.

3. Unlike traditional neural networks that require monotonically increasing odd functions to achieve relatively good performance, this work establishes broader conditions for activation function admissibility, thereby expanding the design space for neural solvers.

4. Experimental results demonstrate that MINN not only solves TVQP problems with improved convergence and robustness but also generalizes effectively to related tasks such as solving the Sylvester equation. Moreover, in the simulation section, the effects of coupling topology parameters on network convergence is explored in detail. In addition, a comparison of robotics experiments illustrates that MINN has a higher control accuracy ($10^{-7}$ m) than the VPRNN ($10^{-6}$ m). This indicates that MINN can be successfully extended to practical applications.

## Results

In this section, the problem description, methodology, and experimental results are given in details.

### TVQP problem description

The TVQP problem is characterized by time-varying nature of its parameters. Below is a concise formulation and its reformulation as a

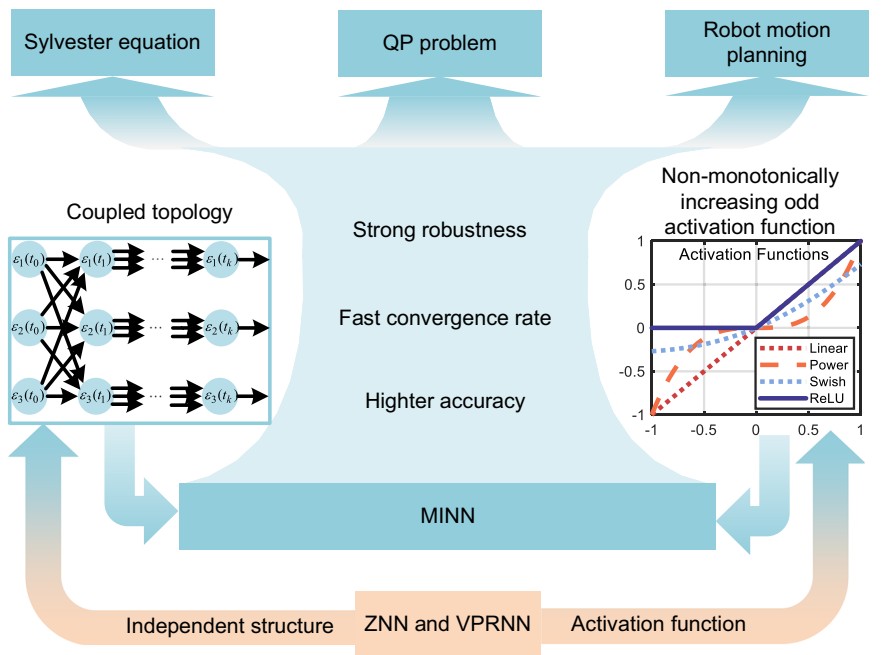

**Fig. 1 | The framework of this paper.** The lower orange part represents the limitations of conventional approaches: ZNN and VPRNN rely on independent structures (where neurons operate in isolation, lacking coordinated interaction) and restrict activation functions to a narrow scope (e.g., only supporting monotonically increasing odd functions). In contrast, MINN introduces two innovations. On the left, the coupled topology depicts dynamic interactions among neurons, overcoming the isolated working limitations of ZNN/VPRNN and enabling synergistic information exchange. On the right, the non-monotonically increasing odd activation functions (e.g., Power, Swish, with Linear and ReLU for comparison)

demonstrates MINN's expanded activation function range, breaking traditional models' strict constraints and accommodating more flexible, practically valuable function forms. The central blue area highlights the advantages of the MINN algorithm: strong robustness, fast convergence rate, and higher computational accuracy. The top part further demonstrates the application of MINN in time-varying tasks: solving the Sylvester equation, QP problems, and robot motion planning. All these scenarios demand real-time, precise, and robust numerical solutions.

time-varying error dynamics equation. The general form of the TVQP problem is

$$\begin{aligned}
\text{min.} \quad & \mathbf{m}^{\mathrm{T}}(t)\mathbf{A}(t)\mathbf{m}(t)/2 + \mathbf{B}^{\mathrm{T}}(t)\mathbf{m}(t) \\
\text{s. t.} \quad & \mathbf{C}(t)\mathbf{m}(t) = \mathbf{D}(t) \\
& \mathbf{P}(t)\mathbf{m}(t) \leq \mathbf{Q}(t)
\end{aligned} \quad (1)$$

where $\mathbf{m}(t) = [m_1(t); m_2(t); \cdots; m_v(t)]$ is the state vector, $\mathbf{A}(t) \in \mathbb{R}^{v \times v}$ is a symmetric matrix, $\mathbf{B}(t) \in \mathbb{R}^v$, $\mathbf{D}(t) \in \mathbb{R}^u$ and $\mathbf{Q}(t) \in \mathbb{R}^l$ time-varying differentiable coefficient vectors, $\mathbf{C}(t) \in \mathbb{R}^{u \times v}$ and $\mathbf{P}(t) \in \mathbb{R}^{l \times v}$ are full row rank matrices.

Using KKT condition and FB function transformation (see Supplementary Note 4 in Supplementary Information), Eq. (1) can be rewritten as

$$\begin{cases}
\mathbf{A}(t)\mathbf{m}^*(t) + \mathbf{B}(t) + \mathbf{C}^{\mathrm{T}}(t)\boldsymbol{\lambda}^*(t) + \mathbf{P}^{\mathrm{T}}(t)\boldsymbol{\mu}^*(t) = 0, \\
\mathbf{C}(t)\mathbf{m}^*(t) - \mathbf{D}(t) = 0, \\
\aleph^{\delta}_{\mathrm{FB}}(\boldsymbol{\phi}(t), \boldsymbol{\mu}^*(t)) = 0.
\end{cases} \quad (2)$$

where $\aleph^{\delta}_{\mathrm{FB}}(\boldsymbol{\phi}(t), \boldsymbol{\mu}^*(t)) = \boldsymbol{\phi}(t) + \boldsymbol{\mu}^*(t) - \sqrt{\boldsymbol{\phi}(t) \circ \boldsymbol{\phi}(t) + \boldsymbol{\mu}^*(t) \circ \boldsymbol{\mu}^*(t) + \delta}$, $\delta \to 0_+$ is a small constant, $\circ$ is Hadamard product, $\boldsymbol{\phi}(t) = \mathbf{Q}(t) - \mathbf{P}(t)\mathbf{m}^*(t)$ and $\mathbf{m}^*(t)$ is optimal solution for TVQP problem (1), $\boldsymbol{\lambda}^*(t) \in \mathbb{R}^u$ and $\boldsymbol{\mu}^*(t) \in \mathbb{R}^l$ are optimal solution for Lagrangian multipliers $\boldsymbol{\lambda}(t)$ and $\boldsymbol{\mu}(t)$.

Based on the above analysis, solving the optimal solution of the TVQP problem (1) has been transformed into solving the solution of (2).

To this end, define the error function $\boldsymbol{\varepsilon}(t)$:

$$\boldsymbol{\varepsilon}(t) = \boldsymbol{\Xi}(t)\mathbf{x}(t) + \boldsymbol{\Phi}(t) \quad (3)$$

where

$$\begin{aligned}
\mathbf{x}(t) &= [\mathbf{m}^{*\mathrm{T}}(t), \boldsymbol{\lambda}^{*\mathrm{T}}(t), \boldsymbol{\mu}^{*\mathrm{T}}(t)]^{\mathrm{T}} \in \mathbb{R}^{v+u+l} \\
\boldsymbol{\Xi}(t) &= \begin{bmatrix} \mathbf{A}(t) & \mathbf{C}^{\mathrm{T}}(t) & \mathbf{P}^{\mathrm{T}}(t) \\ \mathbf{C}(t) & 0 & 0 \\ -\mathbf{P}(t) & 0 & \mathbf{I} \end{bmatrix}, \boldsymbol{\Phi}(t) = \begin{bmatrix} \mathbf{B}(t) \\ -\mathbf{D}(t) \\ \mathbf{Q}(t) - \boldsymbol{\psi}(t) \end{bmatrix} \\
\boldsymbol{\psi}(t) &= \sqrt{\boldsymbol{\phi}(t) \circ \boldsymbol{\phi}(t) + \boldsymbol{\mu}^*(t) \circ \boldsymbol{\mu}^*(t) + \delta}.
\end{aligned}$$

Where $\mathbf{I}$ is the unit matrix of appropriate dimension. The TVQP problem (1) is solved when $\boldsymbol{\varepsilon}(t)$ converges to zero. In the next section, an MINN will be proposed to converge $\boldsymbol{\varepsilon}(t)$.

## The model of meta-interactive neural network
The design formula for the *i*th neuron in MINN is

$$\dot{\varepsilon}_i(t) = -\tau(t)\varepsilon_i(t) + \sum_{j=1}^{v} \alpha_{ij} f_j(\varepsilon_j(t)). \quad (4)$$

Equation (4) comprises two components: $\dot{\varepsilon}_i(t) = -\tau(t)\varepsilon_i(t)$ describes the neuron's intrinsic dynamics, while $\sum_{j=1}^{v} \alpha_{ij} f_j(\varepsilon_j(t))$ denotes the signal received by neuron $i$ from a neighboring neuron, where $\alpha_{ij}$ denotes the gain of neuron $j$ in transmitting the signal to neuron $i$, and $f_j$ is the activation function for that signal, $i, j = \{1, 2, \cdots, v\}$, $v = v + u + l$. Figure 2a shows the structure of *i*th neuron,

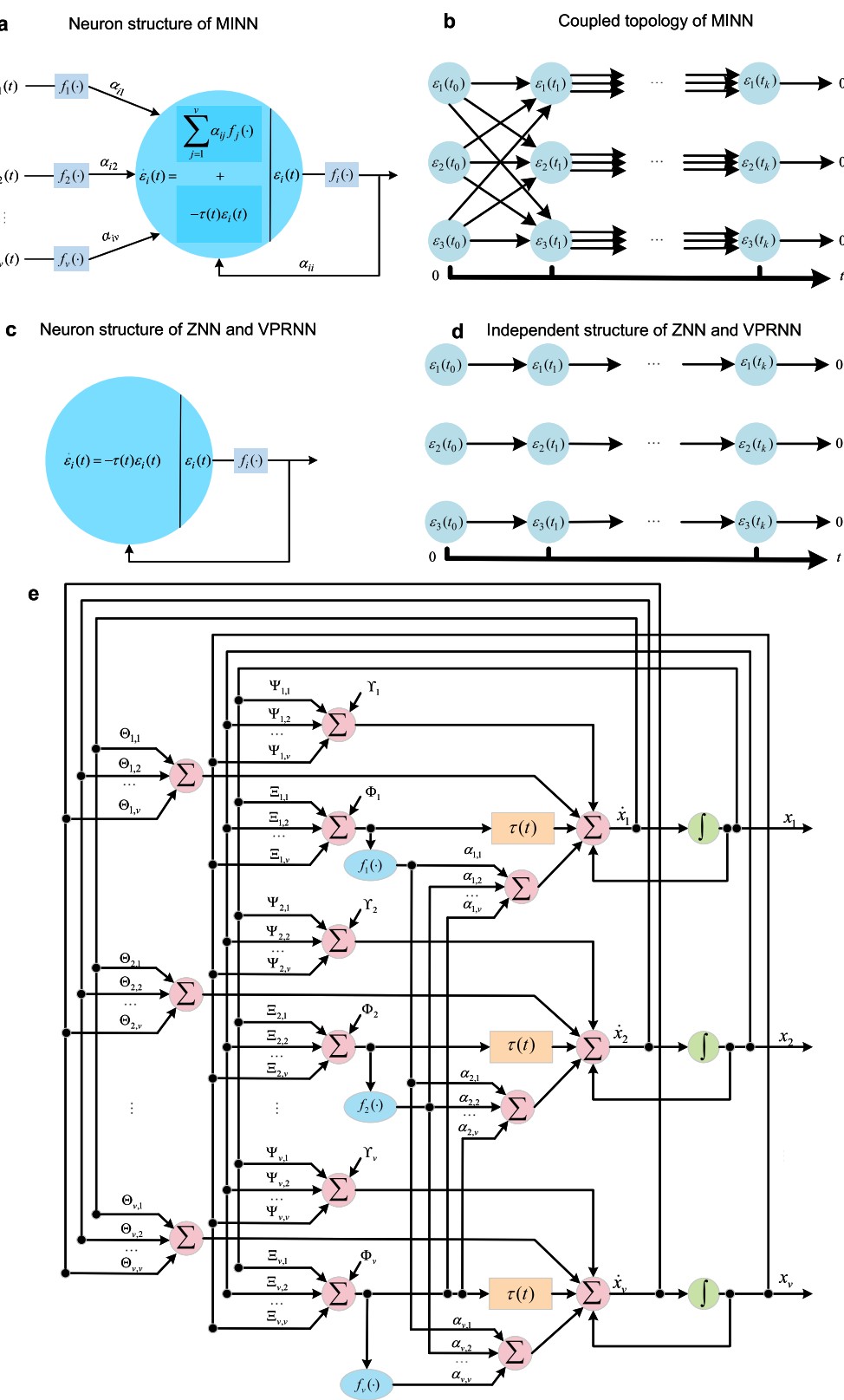

**Fig. 2 | Structure of MINN. a** The structure of the $i$th neuron in MINN. The output of a neuron is transmitted to other neurons via the activation function $f(\cdot)$ and gain $\boldsymbol{\alpha}$. **b** Coupled topology of MINN. Here, we construct a coupled network structure using three neurons as an example. The output of the neurons at time $t_0$ is transmitted to other neurons. **c** The structure of the $i$th neuron in ZNN. **d** The information stream structure of ZNN. **e** Topology diagram of the proposed MINN. Black arrows indicate the direction of data flow. Black dots on the lines represent data exit nodes.

where $\alpha_{ii}$ indicates that the $i$th neuron has autonomous feedback behavior.

Furthermore, for all neurons, we have

$$
\begin{bmatrix} \dot{\varepsilon}_1(t) \\ \dot{\varepsilon}_2(t) \\ \vdots \\ \dot{\varepsilon}_\nu(t) \end{bmatrix} = -\tau(t) \begin{bmatrix} \varepsilon_1(t) \\ \varepsilon_2(t) \\ \vdots \\ \varepsilon_\nu(t) \end{bmatrix} + \begin{bmatrix} \alpha_{11} & \cdots & \alpha_{1j} \\ \alpha_{21} & \cdots & \alpha_{2j} \\ \vdots & \vdots & \vdots \\ \alpha_{i1} & \cdots & \alpha_{ij} \end{bmatrix} \begin{bmatrix} f_1(\varepsilon_1(t)) \\ f_2(\varepsilon_2(t)) \\ \vdots \\ f_j(\varepsilon_j(t)) \end{bmatrix}
$$

and the compact form for the designed formula is

$$
\dot{\boldsymbol{\varepsilon}}(t) = -\tau(t)\boldsymbol{\varepsilon}(t) + \boldsymbol{\alpha}\mathbf{F}(\boldsymbol{\varepsilon}(t)) \tag{5}
$$

where $\dot{\boldsymbol{\varepsilon}}(t) = [\dot{\varepsilon}_1(t), \dot{\varepsilon}_2(t), \ldots, \dot{\varepsilon}_\nu(t)]^\mathsf{T}$, $\mathbf{F}(\cdot) = [f_1, f_2, \ldots, f_\nu]^\mathsf{T}$.

Bringing (3) into (5) yields the implicit dynamical equation for MINN, i.e.,

$$
\boldsymbol{\Theta}(t)\dot{\mathbf{x}}(t) = -\boldsymbol{\Psi}(t)\mathbf{x}(t) - \boldsymbol{\Upsilon}(t) - \tau(t)(\boldsymbol{\Xi}(t)\mathbf{x}(t) + \boldsymbol{\Phi}(t)) + \boldsymbol{\alpha}\mathbf{F}(\boldsymbol{\Xi}(t)\mathbf{x}(t) + \boldsymbol{\Phi}(t)) \tag{6}
$$

where

$$
\boldsymbol{\Theta}(t) = \begin{bmatrix} \mathbf{A}(t) & \mathbf{C}^\mathsf{T}(t) & \mathbf{P}^\mathsf{T}(t) \\ \mathbf{C}(t) & 0 & 0 \\ (\mathbf{M_1}(t) - \mathbf{I})\mathbf{P}(t) & 0 & \mathbf{I} - \mathbf{M_2}(t) \end{bmatrix}
$$

$$
\boldsymbol{\Psi}(t) = \begin{bmatrix} \dot{\mathbf{A}}(t) & \dot{\mathbf{C}}^\mathsf{T}(t) & \dot{\mathbf{P}}^\mathsf{T}(t) \\ \dot{\mathbf{C}}(t) & 0 & 0 \\ (\mathbf{M_1}(t) - \mathbf{I})\dot{\mathbf{P}}(t) & 0 & 0 \end{bmatrix} \boldsymbol{\Upsilon}(t) = \begin{bmatrix} \dot{\mathbf{B}}(t) \\ -\dot{\mathbf{D}}(t) \\ (\mathbf{I} - \mathbf{M_1}(t))\dot{\mathbf{Q}}(t) \end{bmatrix}
$$

$$
\mathbf{M_1}(t) = \mathrm{diag}(\boldsymbol{\phi}(t) \oslash \boldsymbol{\psi}(t)), \mathbf{M_2}(t) = \mathrm{diag}(\boldsymbol{\mu}^*(t) \oslash \boldsymbol{\psi}(t))
$$

and $\oslash$ the Hadamard division. Figure 2e presents the realization diagram of the MINN. Now, the proposed MINN model has been obtained. Next, the differences between MINN and the existing neural network models for solving QP problems will be discussed.

## Differences between MINN and other models

Common neural network for solving TVQP problems include ZNN[11] and VPRNN[12], with the design formula:

$$
\dot{\boldsymbol{\varepsilon}}(t) = -\tau(t)\mathbf{F}(\boldsymbol{\varepsilon}(t)) \tag{7}
$$

when $\tau(t)$ is a time-varying growth parameter, Eq. (7) corresponds to the VPRNN model; when $\tau(t)$ is a constant $\tau$, it becomes the ZNN model. Thus, VPRNN and ZNN share the same structure and characteristics. For convenience, we only compare MINN and ZNN. As can be seen from (5) and (7), there are three differences between MINN and others ZNN.

Difference 1. In ZNN, the dynamic error evolution is typically described by $\boldsymbol{\varepsilon}(t) = e^{-\tau t}$ (assuming that $\mathbf{F}(\cdot)$ is a linear activation function). Obviously, the convergence rate is directly governed by the fixed parameter $\tau$, which limits the network's adaptability. In contrast, MINN employs a time-varying parameter $\tau(t)$. When $\tau(t)$ is a time-varying term that grows over time, such as $\tau(t) = e^t$, the convergence of the system will be significantly improved[12]. Furthermore, when $\tau(t)$ is correlated with the error function, the network has adaptive convergence[18]. Therefore, MINN is a more universal design.

Difference 2. Figure 2b and d illustrate the information flow topologies in MINN and ZNN, where $t_k$ denotes the discrete time index. As time progresses, the final output value of the error expectation is 0. In ZNN, each neuron functions as an isolated processing unit, passing information only to itself across successive time steps. This means that the neuron's output signal at moment $t_k$ can only be used as its own input for the moment $t_{k+1}$. This structure lacks inter-neuronal signal

exchange channels, precluding collaborative capabilities in large-scale networks. In other words, the computational efficiency of multiple neurons is the same as that of one neuron in ZNN, where $\tau(t) = \tau$ is a normal number, i.e., $\varepsilon_i(t) = e^{-\tau t}$ (see Fig. 2c, d). However, with the increase in the number of neurons, we would like to demonstrate the collaborative ability between neurons. In contrast, MINN introduces a coupled architecture, enabling inter-neuronal communication. Figure 2a, b show that the signals of neuron $i$ are influenced by neuron $j$, and a reasonable design allows this coupling relationship to optimize the dynamic process of neuron $i$. In addition, this unique design facilitates scholars to further develop the functions of the network.

Difference 3. ZNN neurons, as information processing nodes, lack intrinsic dynamics. Specifically, without feedback signals, a ZNN neuron's dynamics reduce to $\dot{\boldsymbol{\varepsilon}}(t) = 0$, indicating it merely acts as a data transmission channel without independent information processing capabilities. In contrast, MINN neurons exhibit intrinsic dynamics described by $\dot{\boldsymbol{\varepsilon}}(t) = -\boldsymbol{\tau}(t)\varepsilon(t)$. This inherent dynamic behavior better aligns with the characteristics of biological neurons and enables handling more complex tasks[28–31].

Unlike the network models commonly used in deep learning[32] and machine learning[33], the parameters $\tau(t)$ and matrix $\alpha$ in the proposed MINN (6) do not require training. In other words, MINN employs a training-free learning approach. Once the task requirements and the internal attractor criteria (i.e., the specific state of each neuron) are determined, the weight matrix of the network can be calculated through mathematical procedures. The goal of MINN is to make the neuronal states converge to the expected values through connection weights and mathematical rule. The mathematical basis for this convergence is precisely the Lyapunov stability analysis theorem. The design of connection weights is also determined according to the mathematical rules derived in the next section.

This section compares the proposed MINN with other neural network models in terms of parameter design, topology, and neuron dynamics. Next, we will prove the convergence of MINN and demonstrate its superior convergence mathematically.

## Convergence analysis of MINN under different activation functions

**Theorem 1.** If coupling matrix $\boldsymbol{\alpha}$ is a diagonally dominant matrix with $\alpha_{ii} < 0$, $\alpha_{ij} > 0$, $i \neq j$ and parameter $\tau(t) > 0$. Moreover, the activation function $\mathbf{F}(\cdot)$ satisfies

$$
f_i(\varepsilon_i(t)) = \begin{cases} \geq 0, & \varepsilon_i(t) > 0, \\ = 0, & \varepsilon_i(t) = 0, \\ \leq 0, & \varepsilon_i(t) < 0, \end{cases} \tag{8}
$$

and $f_i(\varepsilon_i(t))$ can be rewritten as

$$
f_i(\varepsilon_i(t)) = \varepsilon_i(t)\varsigma_i(t) \tag{9}
$$

where $\varsigma_i(t) > 0$. Then, for any initial condition $\mathbf{x}(0)$, the proposed MINN (6) guarantees that the error function $\boldsymbol{\varepsilon}(t) = 0$, $t \to +\infty$, i.e., the state vector $\mathbf{x}(t)$ can converge to the unique theoretical solution $\mathbf{x}^*(t) = [\mathbf{m}^{*\mathsf{T}}(t), \boldsymbol{\lambda}^{*\mathsf{T}}(t), \boldsymbol{\mu}^{*\mathsf{T}}(t)]^\mathsf{T}$, whose first $\nu$ elements in $\mathbf{x}^*(t)$ are solutions of TVQP problem (1).

The proof can be found in Supplementary Note 5 in Supplementary Information. Theorem 1 proves that MINN can converge the TVQP problem (1) to its optimal solution, with a detailed mathematical proof. Notably, unlike ZNN and related studies, which assume the activation function is a monotonically increasing odd function, Theorem 1 relaxes the applicability conditions for activation functions. Here, four distinct activation functions (Linear, Power, Swish and ReLU) are employed. All satisfy the requirements in Theorem 1, while Swish and ReLU are not

monotonically increasing odd functions as defined in ZNN. To further explore this aspect, the following corollary is presented.

**Corollary 1.** In the process of solving TVQP Problem (1), the activation function $\mathbf{F}(\cdot)$ in MINN does not have to be restricted to be a monotonically increasing odd function, it can be relaxed to simply satisfy Eqs. (8) and (9).

**Proof.** Recalling (Supplementary Eq. 13) we can see that

$$\dot{V}(t) = -\tau(t)\sum_{i=1}^{\nu}\varepsilon_i^2 + \sum_{i=1}^{\nu}\sum_{j=1}^{\nu}\alpha_{ij}\varsigma_j(t)\varepsilon_i(t)\varepsilon_j(t)$$

it implies that all activation functions can be written in the form (9). The monotonicity and symmetry of the activation function only affect the value of $\varsigma(t)$ (e.g., $\varsigma_{\text{Linear}}(t) = 1$, $\varsigma_{\text{Power}}(t) = \varepsilon^{(\gamma-1)}(t)$). While $\varsigma(t)$ cannot determine whether $\dot{V}(t)$ is less than 0. According to Theorem 1, $\dot{V}(t) < 0$ depends on $\tau(t)$ and $\alpha$. Specifically, as long as $\tau(t) > 0$ and $\alpha$ is a diagonally dominant matrix with $\alpha_{ii} < 0$, $\alpha_{ij} > 0$, $i \neq j$. The MINN is able to make the TVQP problem (1) converge to a theoretical solution as long as the activation function satisfies (8) and (9). This completes the proof.

## Convergence comparisons for MINN and other models

The effectiveness of the proposed MINN for solving the TVQP problem has been detailed in the previous research section. This section focuses on comparing the convergence of the proposed MINN, ZNN and VPRNN.

**Theorem 2.** If the activation function satisfies (8) and (9), then, with the parameter $\tau(t) > 0$, diagonally dominant matrix $\alpha$, where $\alpha_{ii} < 0$, $\alpha_{ij} > 0$, $i \neq j$ and the same initial conditions, MINN (6) has the fastest convergence speed when using the three neural networks MINN, ZNN, VPRNN to solve the TVQP problem (1).

The proof can be found in Supplementary Note 6 in Supplementary Information. The above theoretical results provide a detailed explanation of the superiority of MINN in solving TVQP problems. The core of MINN lies in achieving error function convergence through the design of formula (5). This method is generalizable and applicable to other problems, such as the Sylvester equation. As a key class of matrix equations in linear algebra and matrix theory, the Sylvester equation has widespread applications in control systems, signal processing and numerical analysis. Various methods have been proposed for solving Sylvester equation[34–38]. Among them, refs. 34,35,37 proposed solvers based on ZNN and its improved versions. Compared to the ZNN method, ref. 38 employed VPRNN to enhance convergence efficiency. To demonstrate the generality of MINN, we provide the corresponding experimental setup in the Supplementary Information. Supplementary Fig. 1 shows that MINN can effectively converge Sylvester equations to desired solutions using four activation functions. To further characterize this performance, Supplementary Fig. 2 presents residual of three neural networks solving Sylvester equations under different activation functions, confirming MINN's consistent convergence superiority. Notably, with the non-symmetric ReLU activation function, only MINN achieves error convergence, whereas ZNN and VPRNN cannot. Results in Supplementary Note 1 indicate that MINN exhibits well performance in solving the Sylvester equation.

## Numerical example

In order to verify the authenticity of the above results and to facilitate the reader to reproduce this experiment, we give the detailed experimental parameters. It is worth mentioning that all the experiments in this paper are coded using MATLAB R2021a software and m-files are used to save the code. Select the following parameters for TVQP

problem (1)

$$\mathbf{h}(t) = \begin{bmatrix} h_1(t) \\ h_2(t) \end{bmatrix}, \boldsymbol{\Theta}(t) = \begin{bmatrix} 3+\cos t & \sin t \\ \sin t & 3+\cos t \end{bmatrix},$$

$$\boldsymbol{\Lambda}(t) = \begin{bmatrix} \cos 2t \\ \sin t \end{bmatrix}, \boldsymbol{\Xi}(t) = \begin{bmatrix} \cos 3t & \sin 3t \end{bmatrix}, \boldsymbol{\Psi}(t) = \sin 7t,$$

$$\mathbf{P}(t) = \begin{bmatrix} 1 & 0 & -1 & 0 \\ 0 & 1 & 0 & -1 \end{bmatrix}^{\mathrm{T}}, \mathbf{Q}(t) = \begin{bmatrix} 1.1 & 1.1 & 1.1 & 1.1 \end{bmatrix}^{\mathrm{T}}.$$

Moreover, let $\tau(t) = e^t$ where $e$ is the Euler number, $\mathbf{F}(\cdot)$ assumed to be the four activation functions in (Supplementary Equation 14), and

$$\boldsymbol{\alpha} = \begin{bmatrix} -2.2 & 0.5 & 0.2 & 0.2 & 0.3 & 0.4 & 0.1 \\ 0.5 & -3.1 & 0.4 & 0.8 & 0.2 & 0.1 & 0.3 \\ 0.2 & 0.4 & -4.5 & 0.6 & 0.4 & 0.5 & 0.2 \\ 0.2 & 0.8 & 0.6 & -3.4 & 0.2 & 0.3 & 0.4 \\ 0.3 & 0.2 & 0.4 & 0.2 & -4.2 & 0.5 & 0.7 \\ 0.4 & 0.1 & 0.5 & 0.3 & 0.5 & -5.6 & 0.8 \\ 0.1 & 0.3 & 0.2 & 0.4 & 0.7 & 0.8 & -4.2 \end{bmatrix}$$

Using the above parameters, we constructed MINN and VPRNN solvers. Three random sets of initial conditions are selected, Fig. 3a, b recorded the dynamic trajectories of the state variables in TVQP problem (1) under two solvers with Linear activation function, respectively. Comparison of Fig. 3a, b reveals that the proposed MINN enables $h(t)$ to converge to the theoretical solution, with $h(t)$ strictly confined to the prescribed range. These figures verify the conclusion of Theorem 1.

Figure 4 illustrates the residual convergence process of ZNN, VPRNN and MINN with different activation functions under identical initial conditions, where Fig. 4a is the graph of the four activation functions. Across Fig. 4, MINN residuals with Linear, ReLU, and Swish activation functions are consistently smaller than those of ZNN and VPRNN. Moreover, Fig. 4d provides a detailed view of the residuals' evolution when the Power activation function is used. For the first 0.22 s, the residual of VPRNN is smaller than that of MINN, but a reversal occurs when $\|\varepsilon(t)\|_2 = 1.4032$. Similarly, in the first 0.18s, the residual of ZNN is smaller than that of MINN, and a reversal occurs when $\|\varepsilon(t)\|_2 = 1.63611$. These two results validate the conclusion of Theorem 2. Moreover, Fig. (4) visually demonstrates that MINN converges faster than ZNN and VPRNN under all activation functions. Additionally, the non-monotonically symmetric activation functions (Swish, ReLU) in Fig. (4) also converge the error, which has illustrated the conclusion in Corollary 1. Further, the system can be considered converged when the residual reaches $\|\varepsilon(t)\|_2 = 0.001$. Table 1 shows the times at which $\|\varepsilon(t)\|_2 = 0.001$ and $\|\varepsilon(t)\|_2 = 0.01$ are first reached for three models under the four activation functions. MINN achieves convergence under the Power activation function, while ZNN and VPRNN do not. Moreover, the convergence time for MINN is always shorter than that for ZNN and VPRNN. Additionally, only MINN can achieve convergence in a short time in all cases. Figure 4 and Table 1 further validate the conclusion in Theorem 2.

In addition to system convergence, robustness (resilience to disturbances) is another key metric for evaluating neural network models. For example, in refs. 39–41, the authors studied the robustness of ZNN for solving time-varying problems. In refs. 12,18, the authors discussed the robustness of VPRNN and provided detailed proofs. Excitingly, compared to ZNN and VPRNN, the proposed MINN exhibits the superior robustness when solving TVQP problems. As shown in Supplementary Fig. 3 (see Supplementary Information), under disturbances, MINN achieves a smaller error bound and faster convergence than VPRNN, whereas ZNN fails to converge the system. In summary, compared to widely used ZNN and VPRNN, MINN

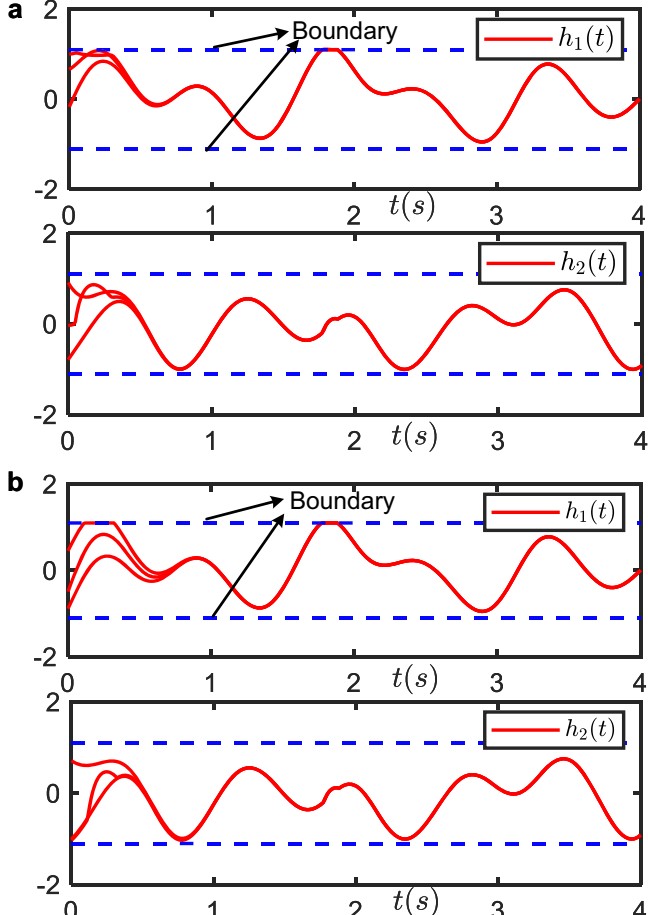

**Fig. 3 | The dynamic trajectories of $h_1(t)$ and $h_2(t)$ under two models with Linear activation function. a** MINN. **b** VPRNN. The solid red line represents the evolution trajectory of the state, while the dashed blue line denotes the inequality constraints in the QP problem. In this experiment, the inequality constraint limits the state value within the range of −1.1 to 1.1. Three sets of random and distinct initial values were selected. When the state value reached the threshold, it is limited to the boundary value, which means the inequality constraint is effective. Under the three different initial conditions, all states converge to the same value over time. The trajectories corresponding to $h_1(t)$ and $h_2(t)$ in both **a**, **b** are identical, confirming that the converged trajectory matches the desired trajectory. The convergence time for states in **a** is significantly faster than that in **b**, demonstrating the superior convergence speed of MINN.

demonstrates well performance in both convergence and robustness. Recent studies have introduced adaptive parameters into network design to enhance convergence performance[18,20]. This approach operates by adjusting the parameter $\tau(t)$, a strategy that can be extended to MINN. Theoretically, MINN will similarly benefit from adaptive parameters design. This will be the direction of our future research. The above analysis shows that compared to ZNN and VPRNN, the MINN is poised to emerge as the optimal choice for future researchers.

It can be seen from (4) that the uniqueness of the MINN is designed by the coupled topology $\sum_{j=1}^{\nu} \alpha_{ij} f_j(\varepsilon_j(t))$ and its own dynamic equation $\dot{\varepsilon}_i(t) = -\tau(t)\varepsilon_i(t)$. It is obviously that the parameters $\alpha_{ij}$ and $\tau(t)$ play a key role in the performance of the MINN. Therefore, the cases of $\tau(t) = 0$ and $\alpha_{ij}$ are discussed as follows.

Case 1: Impact of $\tau(t) = 0$ on the system.

The other parameters are used as in the previous experiments. The residual images of MINN, ZNN and VPRNN are given in Fig. 5a,

where the residuals of VPRNN and ZNN never change. This is because when $\tau(t) = 0$ the design equation in ZNN and VPRNN become $\dot{\varepsilon}(t) = 0$, which means that the rate of change of the error is 0. On the contrary, the MINN still allows the error to converge because $\alpha_{ii}$ instead of $\tau(t)$ acts as a calming effect. Therefore, This case makes the MINN more resistant to interference than the ZNN and VPRNN. Therefore, the above case implies that the MINN is more resistant to interference.

Case 2: When $\tau(t) = 0$, the effect of $\alpha_{ii}$ on the system.

The results of the discussion of Case 1 show that $\alpha_{ii}$ has a calming effect. Therefore, keeping the other parameters constant and gradually changing the value of $\alpha_{ii}$, then, Fig. 5b shows the variation of residuals for MINN. Table 2 counts the convergence time of MINN. Figure 5b and Table 2 show that the convergence is better as $|\alpha_{ii}|$ increases. It can be concluded that the increase of $\tau(t)$ will enhance the convergence effect, which is consistent with the findings in ref. 12.

Case 3: When $\tau(t) = 0$, the effect of $\alpha_{ij}, i \neq j$ on the system.

To explore the role of non-diagonal elements in $\alpha$, redesign the experiment. Let the diagonal element in the matrix $\alpha_{ii} = -4$ and the other elements be kept consistent. The non-diagonal elements $\alpha_{ij}, i \neq j$ are 0.1, 0.2, 0.3, and 0.4 in the four simulations. In addition, two sets of experiments with initial values [2.6, 2.6] and [−2.6, 2.6] were performed, respectively. The results in Fig. 5c show that when the initial values are equally positive, the convergence slows down as $\alpha_{ij}, i \neq j$ increases. Conversely, when the initial values are one positive and one negative i.e., Fig. 5d, the rate of convergence increases as $\alpha_{ij}$ increases. This phenomenon can be explained by exploring Eq. (Supplementary Equation 13), when $\tau(t)$ and $f(\cdot)$ is a linear activation function, Eq. (Supplementary Equation 13) can be written as $\dot{V}(t) = \sum_{j=1}^{\nu} \alpha_{ij}\varsigma_i(t)\varepsilon_i(t)\varepsilon_j(t)$. Both positive initial values imply that $\varepsilon_i(t) \varepsilon_j(t) > 0$. At this point, $\alpha_{ij}$ has a tendency to impede $\dot{V}(t)$ from being negative, i.e., inhibit the rate of convergence. Conversely, when the initial values are one positive and one negative, $\alpha_{ij}$ has the effect of promoting convergence. In summary, the non-diagonal elements react to the information interaction between the elements. The special design of coupling parameters is expected to enhance the function of the network.

The convergence times of the system under Cases 2 and 3 are recorded in Table 2. Additionally, an extra case is recorded, i.e., $\tau(t) = -4$, $\alpha_{ii} = 0$. In this case, the convergence time of the system is consistent with Case 3, as when $\tau(t) = -4$, $\tau(t)$ can serve as a diagonal element $\alpha_{ii}$. Therefore, it is evident that when $\tau(t) = e^t$ and $\alpha_{ii} = 0$, the system will achieve better convergence performance. When using it in practice, it is possible to consider whether to retain the value of the diagonal element based on specific needs. In summary, this example presents various parameter combinations and provides a detailed explanation of the underlying logic by which parameters influence convergence time. This will serve as a valuable resource for further research by subsequent scholars.

## Robotic experiment

Motion planning for robots is a common TVQP problem[13–15]. The real-time angles of each joint angle are calculated to ensure the realization of the end trajectory tracking task. Here, the UFACTORY xArm 6 robot is used as an experimental subject, then, two solvers (MINN and VPRNN) are applied to solve it, respectively. Finally, the results are compared.

The forward kinematics equation for the robot is

$$\mathbf{z}(t) = g(\boldsymbol{\theta}(t)) \qquad (10)$$

where $\mathbf{z}(t)$ is the desired trajectory of the end of the robot, $\boldsymbol{\theta}(t) \in \mathbb{R}^{\nu}$ indicates the $\nu$-dimensional joint angle of the robot, and $\dot{\boldsymbol{\theta}}(t)$ is the angular velocity. $g(\cdot) : \mathbb{R}^{\nu} \to \mathbb{R}^u$ is the mapping relationship from robot joint space to workspace. To solve this problem, the linear

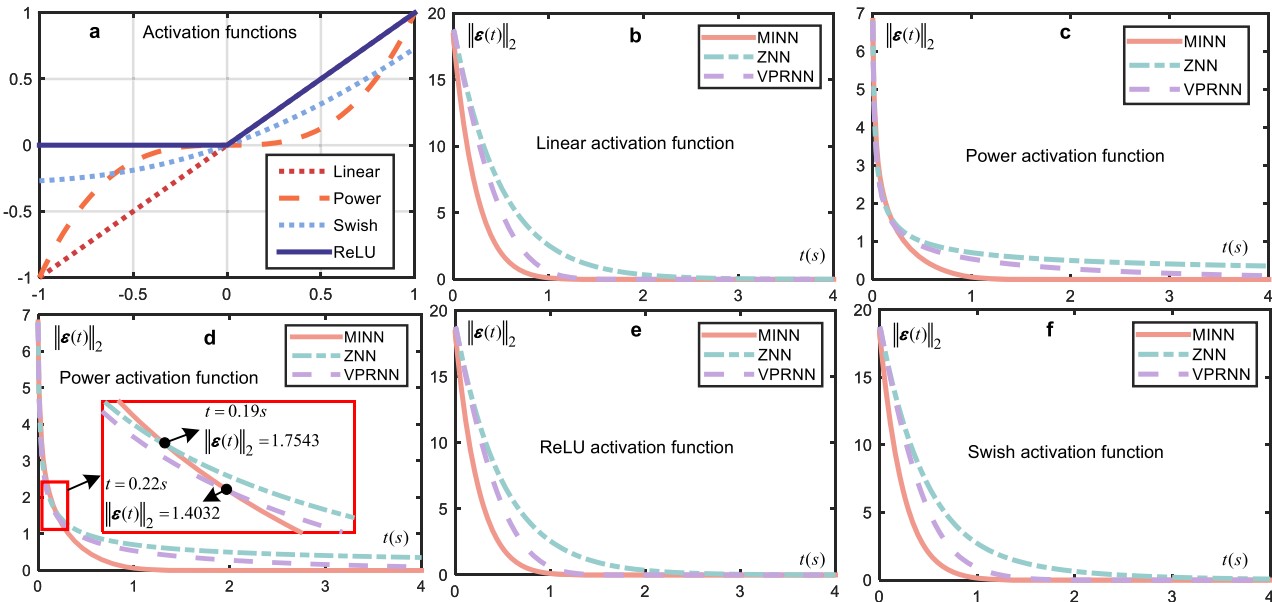

**Fig. 4 | Comparison of residuals. a** Graphs of four activation functions. Residuals of the three models under the Linear activation function (**b**), Power activation function (**c**). ReLU activation function (**e**) and Swish activation function (**f**). **d** Detailed graph of VPRNN ZNN and MINN residual variations under Power activation function with the same initial conditions.

**Table 1 | Time of first achieve $\|\boldsymbol{\varepsilon}(t)\|_2 = 0.01$ and $\|\boldsymbol{\varepsilon}(t)\|_2 = 0.001$ for the three models under different activation functions with same experimental conditions**

|  |  | Linear | Power | Swish | ReLU |
|---|---|---|---|---|---|
| $\|\boldsymbol{\varepsilon}(t)\|_2 = 0.01$ | MINN | 1.32 s | 1.3 s | 1.38 s | 1.32 s |
|  | VPRNN | 1.58 s | can't | 1.95 s | 1.58 s |
|  | ZNN | 3.79 s | can't | can,t | 3.79 s |
| $\|\boldsymbol{\varepsilon}(t)\|_2 = 0.001$ | MINN | 1.54 s | 1.58 s | 1.62 s | 1.54 s |
|  | VPRNN | 1.79 s | can't | 2.23 s | 1.79 s |
|  | ZNN | can't | can't | can't | can't |

equation can be obtained by deriving (10) along time, i.e.,

$$J(\boldsymbol{\theta}(t))(\dot{\boldsymbol{\theta}}(t)) = \dot{\mathbf{z}}(t) \tag{11}$$

where $J(\boldsymbol{\theta}(t)) = \partial \mathbf{z}(t)/\partial \boldsymbol{\theta}(t)$ is the Jacobian matrix. The joint angular velocities of the individual joints at each moment can be obtained by solving the linear equation (11). With the known joint angles of the robot at the initial moment, the joint angles at each moment can be calculated by the difference equation[13]. Moreover, a more efficient research framework that describes the motion planning problem of a robot as a TVQP problem has been proposed in the ref. 42, that is

$$
\begin{aligned}
\min. \quad & \dot{\boldsymbol{\theta}}^{\mathrm{T}}(t)\dot{\boldsymbol{\theta}}(t)/2 \\
\text{s.t.} \quad & J(\boldsymbol{\theta}(t))(\dot{\boldsymbol{\theta}}(t)) = \dot{\mathbf{z}}(t) \\
& \mathbf{H}\dot{\boldsymbol{\theta}}(t) \leq \mathbf{K}
\end{aligned} \tag{12}
$$

where **H** and **K** are physical limit constraints on the joints. In practical robot control, in addition to the primary task of accomplishing the tracking of the desired trajectory, there are some secondary tasks that need to be taken into account, such as the joints of the robot having physical limit constraints, which are the inequalities in (12). In this case,

the TVQP problem (12) is constructed, where the energy minimization of the angular velocity of the joints is taken as the objective function, and the robot kinematic equations and the physical limits of the joints are constraints, respectively. Then, the unified planning of robot multitasking can be accomplished by solving the TVQP problem. Obviously, Eq. (12) is consistent with the TVQP problem (1) described in this paper. Therefore, (12) can be solved with neural networks. The parameters of the robotics experiment are given in the Supplementary Information. Then, Fig. 6a shows the motion of the robot tracking the desired trajectory, where the red five-pointed star is the desired trajectory. Figure 6b gives the position error of the robot tracking a stationary trajectory under two neural network solvers, respectively. Figure 6b shows that the tracking error of the robot end trajectory is at $10^{-7}$ m when using the MINN solver, which is less than the $10^{-6}$ m of the VPRNN. It further shows that the proposed MINN has convergence performance.

## Discussion

In the process of solving TVQP problems, computational speed and stability are two fundamental principles. Previous studies have explored computational methods using neural network models, with ZNN[11] and VPRNN[12] being proposed successively, and their convergence and robustness have been demonstrated. However, through an in-depth analysis of the current mainstream ZNN and VPRNN structures, we found that the independent operation of neurons has certain limitations. Fortunately, inspired by the coupling mechanism of neurons in the human brain, this study proposes the MINN. Our work breaks through the limitation of non-interactive neurons in traditional networks and thoroughly explores the impact of model parameters on system performance. Additionally, MINN expands the scope of activation function usage, providing other perspective for future improvements in neural network models.

The experimental results are encouraging. When solving TVQP problems, MINN demonstrates faster convergence speed and robustness compared to ZNN and VPRNN. Furthermore, MINN exhibits exceptional convergence capability when solving Sylvester equation

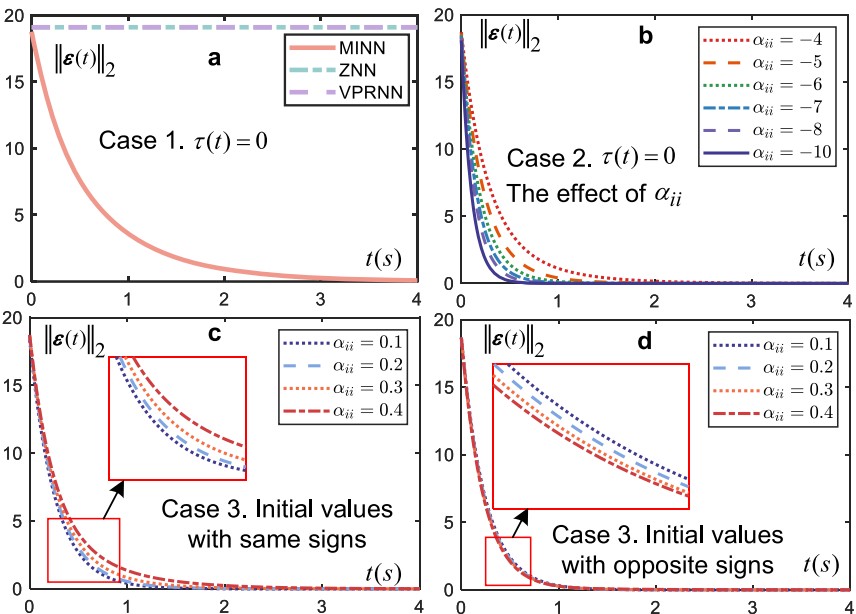

**Fig. 5 | Discussion on coupling parameters. a** Case 1. Residuals of two neural networks with the same initial conditions and $\tau(t) = 0$ under Linear activation function. **b** Case 2. when $\tau(t) = 0$, using the linear activation function, the residuals of MINN under different $\alpha_{ii}$. **c** Case 3. when $\tau(t) = 0$ and initial values are same signs,

the residuals of MINN under different $\alpha_{ii}$ and linear activation function. **d** Case 3. when $\tau(t) = 0$ and initial values are opposite signs, the residuals of MINN under different $\alpha_{ii}$ and linear activation function.

**Table 2 | Time taken for $\|\varepsilon(t)\|_2 = 0.01$ and $\|\varepsilon(t)\|_2 = 0.001$ under linear activation function for five cases**

|  |  | $\alpha_{ii}$ |  |  |  |  |  |
|---|---|---|---|---|---|---|---|
|  |  | -4 | -5 | -6 | -7 | -8 | -10 |
| $\tau(t) = 0$ | $\|\varepsilon(t)\|_2 = 0.01$ | 3.69s | 2.31s | 1.7s | 1.35s | 1.13s | 0.86s |
|  | $\|\varepsilon(t)\|_2 = 0.001$ | can't | 3.17s | 2.31s | 1.83s | 1.51s | 1.14s |

|  |  | $\alpha_{ij}, i \neq j$ (Initial values with same signs) |  |  |  |
|---|---|---|---|---|---|
|  |  | 0.1 | 0.2 | 0.3 | 0.4 |
| $\tau(t) = 0, \alpha_{ii} = -4$ | $\|\varepsilon(t)\|_2 = 0.01$ | 2s | 2.32s | 2.91s | 3.99s |
|  | $\|\varepsilon(t)\|_2 = 0.001$ | 2.64s | 3.11s | 3.95s | can't |

|  |  | $\alpha_{ij}, i \neq j$ (Initial values with opposite signs) |  |  |  |
|---|---|---|---|---|---|
|  |  | 0.1 | 0.2 | 0.3 | 0.4 |
| $\tau(t) = 0, \alpha_{ii} = -4$ | $\|\varepsilon(t)\|_2 = 0.01$ | 1.85s | 1.83s | 1.8s | 1.75s |
|  | $\|\varepsilon(t)\|_2 = 0.001$ | 2.5s | 2.45s | 2.41s | 2.38s |

|  |  | $\alpha_{ij}, i \neq j$ (Initial values with same signs) |  |  |  |
|---|---|---|---|---|---|
|  |  | 0.1 | 0.2 | 0.3 | 0.4 |
| $\tau(t) = -4, \alpha_{ii} = 0$ | $\|\varepsilon(t)\|_2 = 0.01$ | 2s | 2.32s | 2.91s | 3.99s |
|  | $\|\varepsilon(t)\|_2 = 0.001$ | 2.64s | 3.11s | 3.95s | can't |

|  |  | $\alpha_{ij}, i \neq j$ (Initial values with opposite signs) |  |  |  |
|---|---|---|---|---|---|
|  |  | 0.1 | 0.2 | 0.3 | 0.4 |
| $\tau(t) = -4, \alpha_{ii} = 0$ | $\|\varepsilon(t)\|_2 = 0.01$ | 1.85s | 1.83s | 1.8s | 1.75s |
|  | $\|\varepsilon(t)\|_2 = 0.001$ | 2.5s | 2.45s | 2.41s | 2.38s |

problems. Notably, under the Swish activation function, MINN can converge the system, whereas ZNN and VPRNN cannot. Moreover, MINN achieves higher accuracy in robot motion planning problems.

The MINN proposed in this study provides an efficient computational method for the TVQP problem. In future research, we plan to explore the following areas: (1) Enhancing the convergence and robustness of MINN by analyzing the coupling relationships between neurons. (2) Applying MINN to robot motion planning control to validate its performance in practical applications. (3) Exploring the learning mechanism of $\alpha$ and use reinforcement learning or adaptive control frameworks to dynamically optimize the coupling matrix based on problem characteristics.

## Methods

Parameter settings and model construction. In this study, mathematical models were constructed using computer programs, and results were calculated through iteration. The mathematical model for the TVQP problem is given in Eq. (1). Results were directly computed using the implicit dynamic formula (4) of MINN, and the dynamic processes of variables are illustrated in Fig. 3. In the numerical simulation section, a set of parameters for the TVQP problem was provided. Furthermore, the parameters of MINN could be directly obtained from (4) and the parameters of the TVQP problem, where the coupling matrix $\boldsymbol{\alpha}$, activation function $\mathbf{F}(\cdot)$, and $\tau(t)$ were required to satisfy Theorem 1. For ZNN and VPRNN, the implicit dynamic equations is

$$\boldsymbol{\Theta}(t)\dot{\mathbf{x}}(t) = -\boldsymbol{\Psi}(t)\mathbf{x}(t) - \boldsymbol{\Upsilon}(t) - \tau(t)\mathbf{F}(\boldsymbol{\Xi}(t)\mathbf{x}(t) + \boldsymbol{\Phi}(t)) \quad (13)$$

where $\tau(t)$ denotes a constant (for ZNN) and a time-varying constant (for VPRNN), respectively. Other parameters were consistent with those in (4). Similarly, the model for the Sylvester equation is presented in (Supplementary Equation 1), and the robot model is provided in (Supplementary Equation 6) in Supplementary Information. MATLAB was used as the simulation software in this study. Since the basic structure of MINN was designed on differential equations, the ode15s function was employed in the program. The simulation code has been made publicly available, and specific details can be found in the code[43].

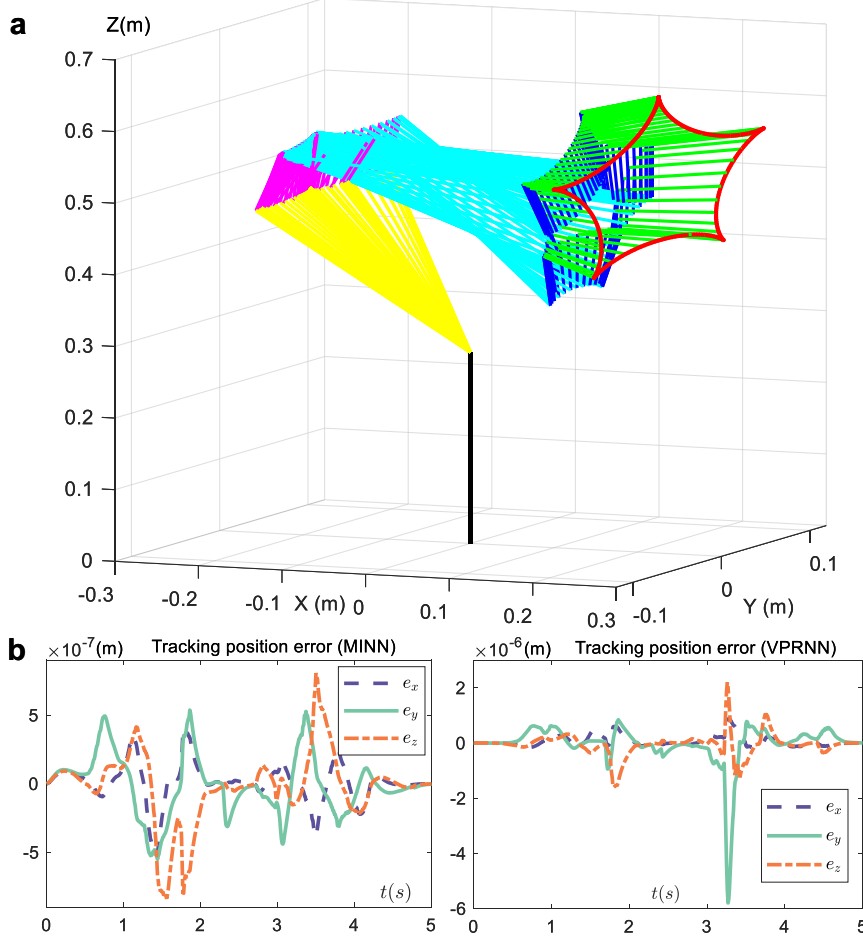

**Fig. 6 | UFACTORY xArm 6 robot experiment. a** Motion state of robot tracking the red desired trajectory. Other colors represent the lengths of each segment of the robot's arm. Joints are represented by connections between different colors. **b** Positional error of robot end-effector tracking desired trajectory under different solvers. $e_x$, $e_y$, and $e_y$ represent errors in different directions within three-dimensional space.

## Code availability
All code accompanying this manuscript is publicly available[43].

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

## Acknowledgements

Z.Z. acknowledges the support of the National Natural Science Foundation under Grants 62373157, the part of the National High-Level Talents Special Support Program (Youth Talent of Technological Innovation of Ten-Thousands Talents Program) under Grant A4220130, International Scientific Research Cooperation Project of Guangdong Science and Technology Plan under Grant 2023A0505050083, the part of Guangzhou Science and Technology Elite Leading Project under Grant 2024A04J5279, the part of the Pazhou Lab Young Scholar Program, the part of the National Key Research and Development Program of China under Grant 2017YFB1002505, Guangdong Soft Science Research Project 2024A1010030001.

## Author contributions

Z.Z. and X.S. designed the research project. X.S. developed the mathematical formalism and performed the simulations. Y.-Q.L. and Y.-M.L. jointly supervised the research. All authors jointly wrote and reviewed the manuscript.

## Competing interests

The authors declare no competing interests.
