## [Transparent Peer Review file · Nature Communications]

A Meta-Interactive Neural Network for Solving Time-Varying Quadratic Programming Problems

Corresponding Author: Professor Zhijun Zhang

Version 0:

Reviewer comments:

Reviewer #1

(Remarks to the Author)

The authors designed and proposed a meta-interactive neural network based on the coupling mechanism of neurons in the human brain, and it has been used to solve the classical quadratic programming problem in mathematics. This design is interesting, and experimental results respond to the effectiveness of this design and show excellent results in robotic motion planning control. Therefore, the reviewer believes that this approach is efficient and inspiring, and it will provide valuable references for subsequent research. The manuscript is well developed with clear theory and results. Therefore, the reviewer suggests a minor revision. The authors should revise the manuscript with the following comments:

1. What problems can be solved by the proposed network other than for solving quadratic programming problems? In other words, the authors should show the potential value of the study.
2. The reviewer would like to suggest the authors compare, if possible, their results with some recent published work and clearly show the new design features in the current work. The following recent relevant references should be discussed:
 - 1) 10.1109/TNNLS.2023.3315332
 - 2) 10.1016/j.neunet.2023.11.058
3. In the experimental section, the authors discuss the effect of parameters $\tau(t)$ and α_{ii} on convergence. The reviewers believe that a more comprehensive discussion is necessary. For example, the significance of the non-diagonal parameters in α .
4. It is suggested that the authors clarify the link between robot motion planning control and the quadratic programming problem.
5. In the robotics experiment, the authors should have given more detailed experimental data.
6. Typos and grammar errors should be eliminated in the work. Please check the whole paper carefully.

Reviewer #2

(Remarks to the Author)

General Assessment:

The manuscript addresses the time-varying quadratic programming (TVQP) problem and proposes a novel Meta-Interactive Neural Network (MINN) model. This model introduces a coupled neuron topology to facilitate intra-network information exchange and employs group dynamics to enhance convergence speed. While the idea is interesting and potentially impactful, the overall quality of the manuscript is currently average, and substantial revisions are necessary before it can be considered for publication.

Detailed Comments:

Language and Clarity:

The manuscript requires thorough proofreading to correct numerous syntax and grammatical errors. Improving the language will significantly enhance readability and clarity.

Example issues include inconsistent spacing, misspellings, and awkward phrasing, particularly in the abstract and introduction.

Software and Implementation Details:

Please explicitly mention the software and computational tools (e.g., MATLAB, Python, TensorFlow, etc.) used for simulation and validation of the proposed MINN model. This information is essential for reproducibility.

Comparative Advantage and Justification:

The manuscript lacks a clear explanation of the advantages of the proposed algorithm over existing methods.

You must justify why a researcher would choose MINN instead of traditional ZNN, VPRNN, or other state-of-the-art approaches.

To support this, please include comparative results with at least two well-established baseline methods. Metrics such as convergence rate, robustness to noise, or computational efficiency should be reported.

(Note: There is no number 4 in the original list; if this was accidental, renumbering may be needed.)

Conclusion Section:

The conclusion needs to be rewritten to better summarize the main contributions, highlight the novelty, and suggest future directions.

Avoid repeating content from earlier sections and aim to emphasize the impact and practical significance of the proposed work.

Literature Review Enhancement:

To provide a more comprehensive background and better contextualize the contribution, please consider incorporating the following relevant references:

- 1) Gao, Y., Tang, Z., Ke, Y., & Stanimirović, P. S. (2024). New activation functions and Zhangians in zeroing neural network and applications to time-varying matrix pseudoinversion. *Mathematics and Computers in Simulation*, 225, 1–12.
- 2) Gerontitis, D., & Tzekis, P. (2024). Solving the generalized Sylvester equation with a novel fast extended neurodynamics. *Numerical Algebra, Control and Optimization*.
- 3) Xie, M., Wang, Q. W., & Chen, J. (2025). Fixed-time TGNN model with the nonlinear activation function for online solution of Sylvester tensor equation. *Numerical Algorithms*.
- 4) Gerontitis, D., Behera, R., Shi, Y., & Stanimirović, P. S. (2022). A robust noise tolerant zeroing neural network for solving time-varying linear matrix equations. *Neurocomputing*, 508, 254–274.
- 5) Yang, Y., Wu, P., Katsikis, V. N., Li, S., & Feng, W. (2025). A novel real-time noise resilient zeroing neural network and its applications to matrix problem solving. *Mathematics and Computers in Simulation*.
- 6) Gerontitis, D., & Tzekis, P. (2025). A New General Fast Neurodynamics (GFN) for Solving Complex Generalized Sylvester Equation With Power Systems Application. *Mathematical Methods in the Applied Sciences*.

Final Recommendation: Major Revision

Version 1:

Reviewer comments:

Reviewer #1

(Remarks to the Author)

After revision, the derivation in the paper is reliable and the simulation results are rich.

The authors provided the detailed comparison between MINN and ZNN in the revised manuscript, which demonstrated the potential influence of MINN.

Moreover, in the robot experiment part, the connection between robot motion planning control and quadratic programming problems has been described in detail.

Generally speaking, this paper is technically sound and somewhat innovative. The reviewer considers that the manuscript

can be accepted.

(Remarks on code availability)

Reviewer #2

(Remarks to the Author)

The manuscript can be accepted in this form.

(Remarks on code availability)

Reviewer #3

(Remarks to the Author)

Dear authors,

The manuscript proposes a new method to solve time-varying quadratic programs by means of a non-linear dynamics. The paper contribution is interesting and could be useful for people working in the field. I have the following comments.

1. First of all, I think that the paper contribution is too narrow and not of significant advance to grant the paper an acceptance to Nature Communications. The aim and scope of the journal is to represent important advances of significance to specialists within each field. I think the paper is better suited for specialist journals, such as IEEE Transactions of Automatic Control, rather than Nature.

2. The English is quite poor and there are many typos (e.g., upper case and lower case letters are used here and there). Some sentences are also awkward. The authors are advised to perform a major polishing effort.

3. On the scientific level, the paper is packaged as proposing a new neural network that can track the solution trajectory of a time-varying optimization problem. Parallelism with nature and interactions are presented. New names invented (Meta-interactive). However, from a basic standpoint, the authors propose a new continuous-time control law to reduce the error of time-varying KKT conditions.

3a. There is no "neural network", in the sense that there is no training of the network. The "neural network" is designed such that the Lyapunov function decreases. As done in control theory. The neural network is just another non-linear control law.

3b. The "interaction" is rather natural; there is no reason to assume that a diagonal control law would work.

I would advise the authors to reframe the contribution to what it represents, especially in the view of a submission to a more specialised journal.

4. When one observe the contribution as it really is, then there is much literature that is missing and methods to compare to.

4a. The work of Wei Ren in this domain is quite extensive (both quadratic and non, both distributed and centralised, both unconstrained and constrained). For example,

"Distributed Continuous-Time Algorithms for Optimal Resource Allocation With Time-Varying Quadratic Cost Functions", IEEE Transactions of Automatic Control, 2021

"Distributed continuous-time time-varying optimization for networked Lagrangian systems with quadratic cost functions", Automatica, 2025

But there are many others.

5. Some interesting avenues for research are left unexplored. For instance, in the spirit of a neural network, it would be interesting to learn the coupling matrix alpha, instead of imposing it.

(Remarks on code availability)

Version 2:

Reviewer comments:

Reviewer #3

(Remarks to the Author)

Dear authors. Thanks for the revised version and the attention you have put in replying to my comments.

I appreciate the effort and the contribution of your paper, but I do not think it is Nature material. I think it should be going through the scrutiny of a specialised journal, where the maths and the contribution to the neural network community can be thoroughly checked.

Time-varying quadratic programming, as you have mentioned, is a topic of growing interest; many methods have been proposed -- yours, while interesting, is not a disruptive advance that would make a difference in the broad community.

(Remarks on code availability)

--

Reviewer #4

(Remarks to the Author)

All the comments have been addressed. The idea is very novel, the theory of the paper is rigorous and the experiments are solid. It makes significant contributions to a wide range of fields. I suggest it be accepted for publication directly.

(Remarks on code availability)

Reviewer #5

(Remarks to the Author)

The paper has been carefully revised in response to the reviewers' concerns and is now ready for acceptance.

(Remarks on code availability)

No

Summary of Changes

Revision Information (#Ref. No. NCOMMS-25-27314)

“A Meta-Interactive Neural Network for Solving Time-Varying Quadratic Programming Problems”

First of all, we (the authors) would like to express our sincere gratitude to the editors and reviewers for their time, efforts and recognition given to our manuscript NCOMMS-25-27314 entitled “A Meta-Interactive Neural Network for Solving Time-Varying Quadratic Programming Problems” which was submitted to *Nature Communications*. As for the recognition, we repeat it as follows: “*This design is **interesting**, and experimental results respond to the effectiveness of this design and show **excellent** results in robotic motion planning control.*”, “*it will provide **valuable** references for subsequent research.*”. “*The manuscript is **well developed** with clear theory and results.*” (Reviewer 1). “*the idea is **interesting** and **potentially impactful***” (Reviewer 2). Thanks a lot and best regards.

Secondly, it is worth pointing out that the editors’ and reviewers’ comments and suggestions have really constructively helped us improve the quality and presentation of our manuscript further. In light of their inspiring comments and suggestions, we have revised our manuscript duly and carefully, with the main revisions listed as follows.

- 1) A Supplementary Information has been provided, including comparisons of different models for solving the Sylvester equation, the robustness of different models for solving TVQP problems, and details of robot experiments. The supplementary section has provided more comparative experiments to support this study.
- 2) According to the editorial requirements, the structure of the manuscript has been adjusted. In particular, the figures have been reorganized and recolored. Some sections have been adjusted. For example, the Conclusion section of the original manuscript has been deleted, and a Discussion section has been added to the revised version. Additionally, the Abstract section has been rewritten.
- 3) As suggested by Reviewer 1, In the robotics experiment section, the relationship between robot motion planning control and quadratic programming problems has been explained in detail.
- 4) As suggested by Reviewer 1, the role of the non-diagonal elements in α has been discussed in detail. Moreover, Fig.6c and Fig.6d has been given.
- 5) As suggested by Reviewer 2, Experiments on the robustness and convergence of MINN, ZNN, and VPRNN under the same conditions have been provided, and the advantages and characteristics of MINN have been explained. In addition, it has been emphasized that the latest algorithms based on ZNN and VPRNN are also applicable to MINN.
- 6) As suggested by Reviewers 1 and 2, we have polished the manuscript. Additionally, references [15], [31], [32], [33], [34], [36], [37] [38] and [39] has been added to the revised version to enrich the theoretical basis of this study. Moreover, according to the editorial suggestions, we have made the research code publicly available.

Finally, we (the authors) would like to thank the editors and the reviewers again sincerely for the inspiring comments provided, and the time and efforts spent in the review.

Responses to Reviewer 1

A Meta-Interactive Neural Network for Solving Time-Varying Quadratic Programming Problems (#NCOMMS-25-27314)

First of all, the authors would like to thank the reviewer for 1) recognizing “*This design is **interesting**, and experimental results respond to the effectiveness of this design and show **excellent** results in robotic motion planning control.*”, “*it will provide **valuable** references for subsequent research.*”. “*The manuscript is **well developed** with clear theory and results.*” and 2) providing many constructive comments to improve further the quality and presentation of this paper.

As suggested by Reviewer 1, We have added Supplementary Note 1 to the Supplementary Information to explain that MINN can also be used to solve the Sylvester equation. In addition, Supplementary Note 3 have provided more details for the robot experiment. In the experiment section, the role of the non-diagonal elements in α has been discussed in detail. Moreover, Fig.6c and Fig.6d has been given. Furthermore, in the robot experiment part, the connection between robot motion planning control and quadratic programming problems has been described in detail. Finally, we have polished the manuscript and added references [31, [36], [39].

Secondly, with many thanks to the reviewer, the authors would like to address his/her comments as below.

Comment 1: *What problems can be solved by the proposed network other than for solving quadratic programming problems? In other words, the authors should show the potential value of the study.*

Response. Thank the reviewer very much for his/her comment. The essence of MINN lies in achieving convergence of the error function through the design of formula (7). This method is general and can be applied to solve other problems, such as the Sylvester equation. To further demonstrate MINN’s potential application capabilities, Supplementary Note 1 is provided in Supplementary Information. Specific details on using MINN to solve the Sylvester equation are provided, and experimental results show that MINN has stronger convergence properties. Moreover, the revised version provides a more detailed explanation, that is

“The above theoretical results provide a detailed explanation of the superiority of MINN in solving TVQP problems. The essence of MINN lies in achieving convergence of the error function through the design of formula (7). This method is general and can be applied to solve other problems, such as the Sylvester equation. As an important class of matrix equations in linear algebra and matrix theory, the Sylvester equation has widespread applications in control systems, signal processing, numerical analysis and so on. Currently, various methods have been proposed to compute the solutions of the Sylvester equation [31-35]. Among them, refs. [31,32,34] proposed solvers based on ZNN and its improved versions. Compared to the ZNN method, ref. [35] employed VPRNN to enhance convergence efficiency. To demonstrate the generality of the proposed MINN, we provide the corresponding experimental setup in the Supplementary Information. Supplementary Fig. 1 demonstrates that MINN can effectively converge Sylvester equations to desired solutions under four activation functions. To further characterize this performance, Supplementary Fig. 2 presents residual of three neural networks solving Sylvester equations under different activation functions, confirming MINN’s consistent convergence superiority. Notably, when the activation function is the non-symmetric ReLU activation function, only MINN can achieve error convergence, while ZNN and VPRNN cannot. The results of Supplementary Note 1 indicate that MINN demonstrates outstanding performance in solving the Sylvester equation.”

“ Consider the smooth time-varying Sylvester equation in ref. [1], [2] as

$$\bar{A}(t)X(t) - X(t)\bar{B}(t) + \bar{C}(t) = 0, t \in [0, +\infty) \quad (\text{Supplementary Equation 1})$$

where t denotes time, $\bar{A} \in \mathbb{R}^{m \times m}$, $\bar{B} \in \mathbb{R}^{n \times n}$ and $\bar{C} \in \mathbb{R}^{m \times n}$ are time-varying smooth coefficient matrices. Without loss of generality, the $\dot{\bar{A}}$, $\dot{\bar{B}}$, $\dot{\bar{C}}$ are the derivative of \bar{A} , \bar{B} , \bar{C} with respect to time. Suppose that the unknown matrix $X \in \mathbb{R}^{m \times n}$ exists, and we are trying to find the unique solution $X^*(t)$ to the Sylvester equation, such that the Sylvester equation holds true. To obtain the unique solution to Sylvester equation (1), a matrix-type error function is defined as

$$\varepsilon(t) = \bar{A}(t)X(t) - X(t)\bar{B}(t) + \bar{C}(t) \quad (\text{Supplementary Equation 2})$$

According to MINN's dynamic design method, the error convergence formula is constructed as

$$\dot{\varepsilon}(t) = -\tau(t)\varepsilon(t) + \alpha F(\varepsilon(t)) \quad (\text{Supplementary Equation 3})$$

where α is a parameter matrix with appropriate dimensions that satisfies the requirements of Theorem 1. Based on the above design formula, the implicit dynamic equation for solving the Sylvester equation under MINN can be obtained as

$$\bar{A}(t)\dot{X}(t) - \dot{X}(t)\bar{B}(t) = -\dot{\bar{A}}(t) + X(t)\dot{\bar{B}}(t) - \dot{\bar{C}}(t) - \tau(t)F(\bar{A}(t)X(t) - X(t)\bar{B}(t) + \bar{C}(t)) + \alpha F(\varepsilon(t)) \quad (\text{Supplementary Equation 4})$$

(Supplementary Equation 4) has provided all the parameters and design details of MINN. Now, we will verify the effectiveness of MINN in solving the Sylvester equation problem. First, let $\tau(t) = 5$ and

$$\alpha = \begin{bmatrix} -4 & 0.3 & 0.2 & 0.5 \\ 0.3 & -4 & 0.1 & 0.6 \\ 0.2 & 0.1 & -4 & 0.4 \\ 0.5 & 0.6 & 0.4 & -4 \end{bmatrix}$$

then, Supplementary Fig. 1 shows the state process of MINN under four activation functions. The blue curve represents the system state, and the red curve represents the desired trajectory. In three experiments with different initial states, all states of the system converged quickly to the desired trajectory. The results in Supplementary Fig. 1 verify the effectiveness of the proposed MINN in solving the Sylvester equation.

Additionally, we present the residual images of MINN, VPRNN, and ZNN under different activation functions. It should be noted that all experiments in Supplementary Fig. 2 were conducted under the same conditions. It can be observed that among the three networks, MINN exhibits the best convergence performance. Furthermore, under the ReLU activation function, only MINN can achieve system convergence. Overall, the results in Supplementary Figs. 1 and 2 confirm that MINN performs well in solving the Sylvester equation problem. In terms of convergence, MINN outperforms the currently popular ZNN and VPRNN.”

Supplementary Fig. 1 The evolution of system state when solving the Sylvester equation using MINN under different activation functions (Linear, Power, ReLU, Swish). The red line represents the expected value, and the blue curve represents the state evolution curve under three different initial values.

Supplementary Fig. 2 The evolution of residuals when solving the Sylvester equation using three models under different activation functions. **a** Linear activation function. **b** Power activation function. **c** ReLU activation function. **d** Swish activation function.

Comment 2: *The reviewer would like to suggest the authors compare, if possible, their results with some recent published work and clearly show the new design features in the current work. The following recent relevant references should be discussed:*

1 [10.1109/TNNLS.2023.3315332](https://arxiv.org/abs/2303.15332)

2 [10.1016/j.neunet.2023.11.058](https://arxiv.org/abs/2311.058)

Response. The authors sincerely thank the reviewer for pointing out such comments which have really inspired the authors to improve the manuscript further. Combining reviewer 2’s comment 5, we added the references suggested by the reviewer as refs. [31] and [36] in the revised manuscript. Additionally, we conducted a detailed comparison with the existing research on MINN. i.e.,

“Currently, various methods have been proposed to compute the solutions of the Sylvester equation [31-35]. Among them, refs. [31,32,34] proposed solvers based on ZNN and its improved versions.” “ In addition to the convergence of the system, robustness (interference resistance) is also an important indicator for evaluating neural network models. For example, in the refs. [36-38], the authors studied the robustness of ZNN for solving time-varying problems. In the refs. [8,14], the authors discussed the robustness of VPRNN and provided detailed proofs.”

Comment 3: *In the experimental section, the authors discuss the effect of parameters $\tau(t)$ and α_{ii} on convergence. The reviewers believe that a more comprehensive discussion is necessary. For example, the significance of the non-*

diagonal parameters in α .

Response. Thank you very much for your careful reading and constructive comments. The author agrees with your point and discusses the impact of non-diagonal elements in α on the network in the revised manuscript. i.e.,

“Case 3: When $\tau(t) = 0$, the effect of $\alpha_{ij}, i \neq j$ on the system.

To explore the role of non-diagonal elements in α , redesign the experiment. Let the diagonal element in the matrix be -4 and the other elements be kept consistent. The diagonal elements are 0.1, 0.2, 0.3 and 0.4 in the four simulations. In addition, two sets of experiments with initial values [2.6, 2.6] and [-2.6, 2.6] were performed, respectively. The results in Fig. 6c show that when the initial values are equally positive, the convergence slows down as $\alpha_{ij}, i \neq j$ increases. Conversely, when the initial values are one positive and one negative i.e. Fig. 6d, the rate of convergence increases as α_{ij} increases. This phenomenon can be explained by exploring Eq. (13), when $\tau(t)$ and $f(\cdot)$ is a linear activation function, Eq. (13) can be written as $\dot{V}(t) = \sum_{j=1}^{\nu} \alpha_{ij} \varsigma_i(t) \varepsilon_i(t) \varepsilon_j(t)$. Both positive initial values imply that $\varepsilon_i(t) \varepsilon_j(t) > 0$. At this point, α_{ij} has a tendency to impede $\dot{V}(t)$ from being negative, i.e., inhibit the rate of convergence. Conversely, when the initial values are one positive and one negative, α_{ij} has the effect of promoting convergence. In summary, the non-diagonal elements react to the information interaction between the elements. The special design of coupling parameters is expected to enhance the function of the network.”

Comment 4: *It is suggested that the authors clarify the link between robot motion planning control and the quadratic programming problem.*

Response. Thank you for your insightful comment. We will clarify the relationship between robot motion planning control and quadratic programming problems from the following two aspects.

1. The compatibility of QP problems and robot motion planning.

QP problems serve as a core optimization tool in robot motion planning, fundamentally transforming physical constraints and task objectives into mathematically efficient optimization problems. For redundant robotic arms (degrees of freedom > 6), inverse kinematics problems have infinitely many solutions, necessitating the use of QP to select the optimal solution from feasible solutions (e.g., minimizing energy consumption, avoiding joint physical limits, etc.). Fortunately, the objective function (quadratic) and constraints (linear/quadratic inequalities) of QP are highly compatible with the core elements of robot motion planning, such as: minimizing joint velocity can be modeled as a quadratic objective function; robot kinematic equations can serve as equality constraints; and joint angle constraints can be represented as linear inequality constraints [a1-a3].

2. Technical contributions of QP to robot motion planning control.

QP can convert multi-objective goals such as trajectory accuracy, energy optimization, and obstacle avoidance into weighted sum forms, while strictly addressing physical constraints through equality/inequality constraints to avoid the suboptimality of traditional heuristic methods. The KKT conditions (Karush-Kuhn-Tucker conditions) of QP provide theoretical guarantees for motion planning, as demonstrated in the literature [a4, a5] demonstrate that the inverse kinematics solutions obtained via QP satisfy stability and convergence requirements .

[a1] Zheng, Xin, Long Jin, and Panfeng Huang. "Distributed collaboration in multimanipulator systems with switching and weight-unbalanced topologies." IEEE/ASME Transactions on Mechatronics 30, 2199-2209 (2024).

[a2] Lian, Yufeng, et al. "Neural dynamics for cooperative motion control of omnidirectional mobile manipulators in the presence of noises: A distributed approach." IEEE/CAA Journal of Automatica Sinica 11, 1605-1620 (2024):.

[a3] Zhang, Zhijun, et al. "Barrier Offset Varying-Parameter Dynamic Learning Network for Solving Dual-Arms Human-Like Behavior Generation." IEEE Transactions on Cognitive and Developmental Systems (2025) DOI: 10.1109/TCDS.2025.3541070.

[a4] Tang, Jinfu, et al. "Robust model predictive control of position and orientation of redundant manipulators." IEEE Transactions on Automation Science and Engineering 22, 16201-16212 (2025).

[a5] Xiao, Lin, et al. "Predefined-time noise immunity ZNN model for dynamic quaternion least squares problem and application to synchronization of hyperchaotic systems." IEEE Transactions on Emerging Topics in Computational Intelligence 8, 1416-1426 (2024).

In summary: QP provides a standardized mapping tool from physical problems to mathematical optimization for motion planning. Its advantages lie in unifying trajectory generation, constraint handling, and optimization objectives within a convex optimization framework, ensuring the optimality and real-time performance of solutions. These ideas are summarized in the revised version. That is

"Motion planning for robots is a common TVQP problem [9-11]. The real-time angles of each joint angle are calculated to ensure the realization of the end trajectory tracking task. Here, the UFACTORY xArm 6 robot is used as an experimental subject, then, two solvers (MINN and VPRNN) are applied to solve it, respectively. Finally, the results are compared.

The forward kinematics equation for the robot is

$$z(t) = g(\theta(t)) \quad (22)$$

where $z(t)$ is the desired trajectory of the end of the robot, $\theta(t) \in \mathbb{R}^v$ indicates the v -dimensional joint angle of the robot, and $\dot{\theta}(t)$ is the angular velocity. $g(\cdot) : \mathbb{R}^v \rightarrow \mathbb{R}^u$ is the mapping relationship from robot joint space to workspace. To solve this problem, the linear equation can be obtained by deriving (22) along time, i.e.,

$$J(\theta(t))(\dot{\theta}(t)) = \dot{z}(t) \quad (23)$$

where $J(\theta(t)) = \partial z(t)/\partial \theta(t)$ is the Jacobian matrix. The joint angular velocities of the individual joints at each moment can be obtained by solving the linear equation (23). With the known joint angles of the robot at the initial moment, the joint angles at each moment can be calculated by the difference equation [9]. Moreover, a more efficient research framework that describes the motion planning problem of a robot as a TVQP problem has been proposed in the ref. [39], that is

$$\begin{aligned} \min. \quad & \dot{\theta}^T(t)\dot{\theta}(t)/2 \\ \text{s.t.} \quad & J(\theta(t))(\dot{\theta}(t)) = \dot{z}(t) \\ & H\dot{\theta} \leq K \end{aligned} \quad (24)$$

where H and K are physical limit constraints on the joints. In practical robot control, in addition to the primary task of accomplishing the tracking of the desired trajectory, there are some secondary tasks that need to be taken into account, such as the joints of the robot having physical limit constraints, which are the inequalities in (24). In this case, the TVQP problem (24) is constructed, where the energy minimization of the angular velocity of the joints is taken as the objective function, and the robot kinematic equations and the physical limits of the joints are constraints, respectively. Then, the unified planning of robot multitasking can be accomplished by solving the TVQP problem. Obviously, Eq. (24) is consistent with the TVQP problem (1) described in this paper. Therefore, (24) can be solved with neural networks. The parameters of the robotics experiment are given in the Supplementary Information. Then, Fig. 7a shows the motion of the robot tracking the desired trajectory, where the red five-pointed star is the desired trajectory. Fig. 7b gives the position error of the robot tracking a stationary trajectory under two neural network solvers, respectively. Fig. 7b shows that the tracking error of the robot end trajectory is at 10^{-7} m

when using the MINN solver, which is less than the 10^{-6} m of the VPRNN. It further shows that the proposed MINN has convergence performance. ”

Comment 5: *In the robotics experiment, the authors should have given more detailed experimental data.*

Response. Thank you very much for your suggestion. In order to enrich the details of the robot experiment, we have provided a detailed explanation in Supplementary Note 3 of Supplementary Information. In particular, we have provided a more detailed description of the joint limit constraints. In addition, Supplementary Fig. 4 shows the trends in the angles and angular velocities of each joint of the robot. That is

“The motion planning problem for UFACTORY xArm 6 robot can be summarized as follows:

$$\begin{aligned} \min. \quad & \dot{\theta}^T(t)\dot{\theta}(t)/2 \\ \text{s.t.} \quad & J(\theta(t))(\dot{\theta}(t)) = \dot{z}(t) \\ & H\dot{\theta} \leq K \end{aligned} \quad (\text{Supplementary Equation 6})$$

where the equality constraints in the constraint equations are the derivatives of the robot’s kinematic equations with respect to time. The inequality constraints are the physical limits of the robot’s joints (angles and angular velocities), and $H = \begin{bmatrix} I & 0 \\ 0 & -I \end{bmatrix}$, $K = \begin{bmatrix} K^+ \\ -K^- \end{bmatrix}$, I is unit matrix. the physical limits of the joints must be taken into account in the motion planning for the manipulators. Excessive angular and angular velocity outputs can cause damage to the manipulators. Therefore the physical limits of the robot is set as

$$\theta \in [\theta^-, \theta^+], \dot{\theta} \in [\dot{\theta}^-, \dot{\theta}^+] \quad (\text{Supplementary Equation 7})$$

where θ_i^- , θ_i^+ , $\dot{\theta}_i^-$ and $\dot{\theta}_i^+$ denote the upper and lower bounds of joint angles and angular velocities, respectively. $\theta(t) = [\theta_1(t), \theta_2(t), \dots, \theta_n(t)]^T$ is the joint vector for the n -dimensional robot manipulator.

According to [4], the constraints on the joints is solved at the velocity level. Thus, (Supplementary Equation 7) is rewritten as

$$\varrho^- = \varrho(\theta^- - \theta) \leq \dot{\theta} \leq \varrho(\theta^+ - \theta) = \varrho^+$$

where ϱ is a specific parameter of manipulator, and the bound constraint is

$$K^- = \max\{\varrho_i^-, \dot{\theta}^-\} \leq \dot{\theta} \leq \min\{\varrho_i^+, \dot{\theta}^+\} = K^+. \quad (\text{Supplementary Equation 8})$$

Then, the physical limit constraints of the joints is obtain as $K^- \leq \dot{\theta} \leq K^+$, where $K^- = [K_1^{-T}, K_2^{-T}, \dots, K_n^{-T}]^T$ and $K^+ = [K_1^{+T}, K_2^{+T}, \dots, K_n^{+T}]^T$. Furthermore, the physical limit constraints on the joints can be summarized as $H\dot{\theta} \leq K$.

Now, we set the initial joint angle of the robotic arm as $[0; -2.3047; 0.8002; 0; -0.3473; 0]$ rad, and all joint angles and angular velocity limits are 2.5rad and 1.2rad/s, respectively. Then, Supplementary Fig. 4 shows the evolution of the angles and angular velocities of each joint of the robot under MINN over time. It can be seen that the physical limits of the robot’s joints are restricted to a predetermined range, and when the angular velocity reaches the threshold, it will remain within but not exceed the set limit value.”

Supplementary Fig. 4 Evolution of joint angles and angular velocities when solving robot motion planning using MINN. **a** joint angles. **b** angular velocities.

Comment 6: *Typos and grammar errors should be eliminated in the work. Please check the whole paper carefully.*

Response. Thank you very much for your suggestions. Based on which we have carefully reviewed the manuscript and polished the grammar and vocabulary to improve its readability. Specific information can be found in the blue-marked sections of the revised draft.

Finally, the authors would like to say thanks again sincerely to the reviewer for his/her time and efforts spent in reviewing the manuscript, as well as many constructive comments he/she has provided which have improved much further the presentation and quality of this manuscript.

Responses to Reviewer 2

A Meta-Interactive Neural Network for Solving Time-Varying Quadratic Programming Problems (#NCOMMS-25-27314)

First of all, the authors would like to thank the reviewer for 1) recognizing the contribution of the paper, i.e., “*the idea is interesting and potentially impactful*”, and 2) providing constructive comments to improve further the quality and presentation of the paper.

As suggested by Reviewer 2, We have demonstrated that the experiments in this study were conducted using MATLAB software, and we are willing to share the code with the reviewers. Additionally, to highlight the advantages of MINN, we have redesigned the experiments. ZNN, VPRNN, and MINN were used as experimental subjects, and the convergence and robustness (Supplementary Note 2 in Supplementary Information) of different models have been compared under the same experimental conditions. Furthermore, Figures 4, 8, and 9 from the original manuscript have been redrawn and restructured as Figure 5 in the revised version. In addition, the data in Table 1 has been modified. The Conclusion section of the original manuscript has been deleted, and a Discussion section has been added to the revised version. Finally, we polished the manuscript and added references [15], [32], [33], [34], [37] and [38].

Secondly, with many thanks to the reviewer, the authors would like to address his/her comments as below.

Comment 1: *Language and Clarity:*

The manuscript requires thorough proofreading to correct numerous syntax and grammatical errors. Improving the language will significantly enhance readability and clarity. Example issues include inconsistent spacing, misspellings, and awkward phrasing, particularly in the abstract and introduction.

Response. Thank the reviewer for his/her suggestions. According to your advice, the author has thoroughly proofread the manuscript and corrected the grammatical errors. In addition, we have asked a professional agency to polish the grammar to improve the readability and clarity of the manuscript. Furthermore, according to the editor’s suggestions, we have rewritten and reformatted the manuscript. The author hopes that our work will satisfy you.

Comment 2: *Software and Implementation Details:*

Please explicitly mention the software and computational tools (e.g., MATLAB, Python, TensorFlow, etc.) used for simulation and validation of the proposed MINN model. This information is essential for reproducibility.

Response. Thank the reviewer for your comment. The environment used for simulation and experimentation in this study was based on MATLAB R2021a software. We wrote the code and saved it as m-files for convenience. Additionally, to facilitate readers’ reproduction of this study, we provided more detailed data in the simulation section of the revised manuscript. Furthermore, we have provided Supplementary Information materials, including three supporting documents and specific details. Of course, All code used in this study is available through <https://github.com/245578/A-Meta-Interactive-Neural-Network-for-Solving-Time-Varying-Quadratic-Programming-Problems.git>.

Comment 3: *Comparative Advantage and Justification:*

The manuscript lacks a clear explanation of the advantages of the proposed algorithm over existing methods. You must justify why a researcher would choose MINN instead of traditional ZNN, VPRNN, or other state-of-the-art approaches. To support this, please include comparative results with at least two well-established baseline methods. Metrics such as convergence rate, robustness to noise, or computational efficiency should be reported. (Note: There is no number 4 in the original list; if this was accidental, renumbering may be needed.)

TABLE I: Time of first achieve $\|\varepsilon(t)\|_2 = 0.01$ and $\|\varepsilon(t)\|_2 = 0.001$ for the three models under different activation functions with same experimental conditions

		Linear	Power	Swish	ReLU
$\ \varepsilon(t)\ _2 = 0.01$	MINN	1.32s	1.3s	1.38s	1.32s
	VPRNN	1.58s	can't	1.95s	1.58s
	ZNN	3.79s	can't	can,t	3.79s
$\ \varepsilon(t)\ _2 = 0.001$	MINN	1.54s	1.58s	1.62s	1.54s
	VPRNN	1.79s	can't	2.23s	1.79s
	ZNN	can't	can't	can't	can't

Response. Thank the reviewer for his/her suggestions. In the original manuscript, we only compared the differences between VPRNN and MINN in terms of convergence. The authors agree with the reviewer's comments that a simple comparison is obviously insufficient. Therefore, in the revised version, we have added ZNN as a comparison object. In the simulation section, we compared the performance of ZNN, VPRNN, and the proposed MINN under the same experimental conditions. Table I was rewritten to record the convergence times of the three models.

Additionally, we compared the robustness of the three models for solving TVQP problems in the Supplementary Information. For the convenience of the reviewers, we present the specific content as follows.

“Ref. [2] describes a neural network model with disturbances. Thus, considering differentiation errors and model-implementation errors, a perturbed model of MINN can be described as

$$\Theta(t)\dot{x}(t) = -(\Psi(t) + \Delta\Psi(t))x(t) - \Upsilon(t) - \tau(t)(\Xi(t)x(t) + \Phi(t)) + \Delta s(t) + \alpha F(\Xi(t)x(t) + \Phi(t)) \quad (\text{Supplementary Equation 5})$$

where $\Delta\Psi(t) \in \mathbb{R}^{\nu \times \nu}$ denotes the differentiation error of coefficient matrix $\Psi(t)$, and $\Delta s(t) \in \mathbb{R}^{\nu}$ denotes the model-implementation error. Setting network parameters $\tau(t) = 2$, the system state is limited to $[-3, 3]$, and

$$\Delta\Psi(t) = \begin{bmatrix} 2\cos(t) & \cos(t) & \sin(t) & \cos(2t) & \sin(t) & 2\cos(t) & \cos(3t) \\ \cos(t) & \sin(2t) & \cos(t) & \cos(2t) & 3\sin(t) & 2\sin(t) & \cos(t) \\ \sin(t) & \cos(t) & \cos(t) & \sin(t) & 2\cos(t) & \cos(3t) & \sin(t) \\ \cos(2t) & \cos(2t) & \sin(t) & \sin(t) & 2\cos(t) & \cos(2t) & 3\cos(t) \\ \sin(t) & 3\sin(t) & 2\cos(t) & 2\cos(t) & 2\cos(t) & \sin(3t) & 2\cos(t) \\ 2\cos(t) & 2\sin(t) & \cos(3t) & \cos(2t) & \sin(3t) & \sin(t) & \cos(t) \\ \cos(3t) & \cos(t) & \sin(t) & 3\cos(t) & 2\cos(t) & \cos(t) & \cos(2t) \end{bmatrix}, \Delta s(t) = \begin{bmatrix} \cos(2t) \\ 2\cos(t) \\ \sin(t) \\ 3\cos(2t) \\ \sin(2t) \\ 2\cos(t) \\ \cos(t) \end{bmatrix}$$

the other parameters are consistent with those in the experimental section of the main text. Supplementary Fig. 3 shows the residuals of ZNN, VPRNN, and MINN when solving the TVQP problem under perturbations using two activation functions. Among them, the linear activation function represents a monotonically increasing activation function type, while the Swish activation function represents a non-monotonically increasing type. Moreover, In Supplementary Fig. 3, ZNN cannot guarantee convergence of residuals under perturbations, while VPRNN can converge the system but at a slower rate than MINN. Overall, MINN exhibits the best robustness and effectively addresses the convergence issue of the TVQP problem under perturbations.”

Supplementary Fig. 3 The evolution of residuals when solving TVQP with perturbations under different activation functions in three models. **a** Linear activation function. **b** Swish activation function.

We have made the corresponding descriptions in the revised draft. That is

“In addition to the convergence of the system, robustness (interference resistance) is also an important indicator for evaluating neural network models. For example, in the refs. [36-38], the authors studied the robustness of ZNN for solving time-varying problems. In the refs. [8,14], the authors discussed the robustness of VPRNN and provided detailed proofs. Excitingly, compared to ZNN and VPRNN, the MINN proposed in this paper exhibits the strongest robustness when solving TVQP problems. As shown in Supplementary Fig. 3 in the Supplementary Information, under the presence of disturbances, MINN has a smaller error bound and faster convergence rate than VPRNN, while ZNN fails to converge the system. In summary, compared to ZNN and VPRNN, which are widely used in existing research, MINN demonstrates outstanding performance in both convergence and robustness. Recent research has introduced adaptive parameters into network design to achieve better convergence results [14,16]. This method is achieved by changing the parameter $\tau(t)$. Therefore, the same method can be used in MINN. Theoretically, MINN will also benefit from the design of adaptive parameters. This will be the direction of our future research. The above analysis shows that compared with ZNN and VPRNN, the MINN is expected to be the optimal choice for subsequent researcher.”

Comment 4: *Conclusion Section:*

The conclusion needs to be rewritten to better summarize the main contributions, highlight the novelty, and suggest future directions. Avoid repeating content from earlier sections and aim to emphasize the impact and practical significance of the proposed work.

Response. Thank you very much for your comment. According to the journal’s formatting requirements, we deleted the “Conclusion” section from the original manuscript and added a “Discussion” section. In the Discussion section, we summarized the work of this paper to better summarize the main contributions and highlight its novelty. In addition, we provided directions for future development. That is

Discussion

In the process of solving TVQP problems, computational speed and stability are two fundamental principles. Previous studies have explored computational methods using neural network models, with ZNN [7] and VPRNN [8] being proposed successively, and their convergence and robustness have been demonstrated. However, through an in-depth analysis of the current mainstream ZNN and VPRNN structures, we found that the independent operation of neurons has certain limitations. Fortunately, inspired by the coupling mechanism of neurons in the human brain,

this study proposes a novel MINN. Our work breaks through the limitation of non-interactive neurons in traditional networks and thoroughly explores the impact of model parameters on system performance. Additionally, MINN expands the scope of activation function usage, providing a new perspective for future improvements in neural network models.

The experimental results are encouraging. When solving TVQP problems, MINN demonstrates faster convergence speed and robustness compared to ZNN and VPRNN. Furthermore, MINN exhibits exceptional convergence capability when solving Sylvester equation problems. Notably, under the Swish activation function, MINN can converge the system, whereas ZNN and VPRNN cannot. Furthermore, MINN achieves higher accuracy in robot motion planning problems.

The MINN proposed in this study provides an efficient computational method for the TVQP problem. In future research, we plan to explore the following areas: 1) Enhancing the convergence and robustness of MINN by analyzing the coupling relationships between neurons. 2) Applying MINN to robot motion planning control to validate its performance in practical applications.”

Comment 5: *Literature Review Enhancement:*

To provide a more comprehensive background and better contextualize the contribution, please consider incorporating the following relevant references:

1) Gao, Y., Tang, Z., Ke, Y., Stanimirović, P. S. (2024). *New activation functions and Zhangians in zeroing neural network and applications to time-varying matrix pseudoinversion. Mathematics and Computers in Simulation, 225, 1-12.*

2) Gerontitis, D., Tzekis, P. (2024). *Solving the generalized Sylvester equation with a novel fast extended neurodynamics. Numerical Algebra, Control and Optimization.*

3) Xie, M., Wang, Q. W., Chen, J. (2025). *Fixed-time TGNN model with the nonlinear activation function for online solution of Sylvester tensor equation. Numerical Algorithms.*

4) Gerontitis, D., Behera, R., Shi, Y., Stanimirović, P. S. (2022). *A robust noise tolerant zeroing neural network for solving time-varying linear matrix equations. Neurocomputing, 508, 254-274.*

5) Yang, Y., Wu, P., Katsikis, V. N., Li, S., Feng, W. (2025). *A novel real-time noise resilient zeroing neural network and its applications to matrix problem solving. Mathematics and Computers in Simulation.*

6) Gerontitis, D., Tzekis, P. (2025). *A New General Fast Neurodynamics (GFN) for Solving Complex Generalized Sylvester Equation With Power Systems Application. Mathematical Methods in the Applied Sciences.*

Response. Thank you very much for your suggestions. The references you mentioned have provided a more comprehensive background of our work. We have added these references as [15], [32], [33], [34], [37], and [38] in the revised version.

Finally, the authors would like to thank the reviewer again and sincerely for his/her time and efforts spent in reviewing the manuscript, as well as many constructive comments he/she has provided which have really helped the authors improve much further the presentation and quality of this manuscript.

Summary of Changes

Revision Information (#Ref. No. NCOMMS-25-27314A)

“A Meta-Interactive Neural Network for Solving Time-Varying Quadratic Programming Problems”

Zhijun Zhang, Xiangliang Sun, Yiqi Liu and Yamei Luo

First of all, we (the authors) would like to express our sincere gratitude to the editors and reviewers for their time, efforts and recognition given to our manuscript NCOMMS-25-27314A entitled “A **Meta-Interactive Neural Network for Solving Time-Varying Quadratic Programming Problems**” which was submitted to *Nature Communications*. As for the recognition, we repeat it as follows: “*the derivation in the paper is **reliable** and the simulation results are **rich**”*, “*The reviewer considers that the manuscript can be **accepted**”*. (Reviewer 1). “*The manuscript can be **accepted** in this form.*” (Reviewer 2). “*The paper contribution is **interesting** and could be **useful** for people working in the field.*” (Reviewer 3). Thanks a lot and best regards.

Secondly, it is worth pointing out that the reviewers’ comments and suggestions have really constructively helped us improve the quality and presentation of our manuscript further. In light of their inspiring comments and suggestions, we have revised our manuscript duly and carefully, with the main revisions listed as follows.

- 1) As suggested by Reviewer 3, we have explained why the proposed method belongs to neural networks and provided the neural network architecture, activation functions and dynamic model in the revised version. In addition, we have clarified why parameters do not need to be trained.
- 2) As suggested by Reviewer 3, expectations for future work have been added to the Discussion section, and we have added a comparison with related works in the Introduction section.
- 3) As suggested by Reviewer 3, we have clarified the significance of the diagonal elements’ impact on the system in the revised version. Furthermore, Table 2 has been redesigned to better compare the impact of parameter changes on the convergence performance of the system.
- 4) As suggested by Reviewer 3, we have polished the manuscript and corrected some typos. In addition, we have asked more than three high level researchers to improve further the manuscript. Additionally, we have compared and discussed the excellent papers recommended by Reviewer 3 and related papers, i.e., references [4], [7], [8], [34] and [35] in the revised manuscript.

Finally, we (the authors) would like to thank the editors and the reviewers again sincerely for the inspiring comments provided, and the time and efforts spent in the review.

Responses to Reviewer 1

Comment: *After revision, the derivation in the paper is **reliable** and the simulation results are **rich**. The authors provided the detailed comparison between MINN and ZNN in the revised manuscript, which demonstrated the potential influence of MINN. Moreover, in the robot experiment part, the connection between robot motion planning control and quadratic programming problems has been described in detail. Generally speaking, this paper is **technically sound** and somewhat **innovative**. The reviewer considers that the manuscript can be **accepted**.*

Response. The authors sincerely thank the reviewer for his/her recognition and praise. The authors have further improved the quality of the paper and submitted the revised manuscript.

Responses to Reviewer 2

Comment: *The manuscript can be **accepted** in this form.*

Response. The authors sincerely thank the reviewer for his/her recognition and acceptance of the manuscript.

Responses to Reviewer 3

Comment 1: *First of all, I think that the paper contribution is too narrow and not of significant advance to grant the paper an acceptance to Nature Communications. The aim and scope of the journal is to represent important advances of significance to specialists within each field. I think the paper is better suited for specialist journals, such as IEEE Transactions of Automatic Control, rather than Nature.*

Response. The authors respect the reviewer, but really disagree with the reviewer's comment on the contribution of the paper. In fact, the significant and unique contribution of the paper is that it is the first time to imitate the information exchange behaviors between biological neurons, and construct a coupled topology for artificial neurons, which enables information exchange within the network, and utilizes group dynamics to accelerate the convergence process. It breaks the inherent pattern of traditional neural networks that can only improve network performance by fine-tuning parameters. Its significance lies in better matching the information exchange patterns of biological nervous systems and revolutionizing the weight update methods commonly used in existing neural networks. Considering the aim and scope of the journal is to represent important advances of significance to specialists within each field, the authors would like to publish the paper in Nature Communications so that a wider audience and more interested researchers may benefit.

The paper contributes to some specific areas as follows.

1. Contribution to Artificial Intelligence

Traditional dynamic neural networks are constrained by rigid activation function requirements (monotonically increasing odd functions), limiting their adaptability to real-world. By contrast, MINN relaxes this constraint (validated in Corollary 1) and introduces a coupled neuronal topology, enabling it to handle non-monotonic, non-symmetric activation functions (e.g., ReLU, Swish) while guaranteeing stable convergence. This advance improves optimization performance and enriches research in the AI field: it bridges the gap between static deep learning models (which rely on flexible activation functions) and dynamic optimization networks (which demand stability), providing a new paradigm for designing real-time, adaptive AI systems (e.g., dynamic decision-making in reinforcement learning or adaptive control in intelligent agents).

2. Contribution to Numerical Computation

MINN has demonstrated exceptional convergence and robustness in solving time-varying problems that are widespread in nature and engineering fields. Experimental results reported in the paper verify that: (1) MINN achieves faster convergence performance compared with traditional neural networks. (2) Stronger robustness. MINN can suppress errors to near-zero levels even in the presence of external interference or internal errors. These properties make MINN a universal tool for time-varying numerical tasks dependent on real-time and reliable computation.

3. Contribution to Robotics

Robotic systems demand high-precision, real-time optimization under physical constraints—requirements that MINN addresses effectively, with additional value for future research. Our experiments show that MINN improves the control accuracy of robotic motion planning from 10^{-6} m (VPRNN) to 10^{-7} m, a 10-fold enhancement critical for precision tasks. Beyond accuracy, coupled neuronal topology of MINN pioneered a new design possibilities for coordination of joints. This opens new avenues for optimizing robotic dynamics. These advantages directly benefit robotic applications in real-world:

- Industrial robotics: High-precision assembly (e.g., semiconductor manufacturing) where sub-micron accuracy is required.
- Collaborative robots: Human-safe motion planning via real-time constraint handling [R1].

- Intelligent robotics: Dynamic obstacle avoidance (e.g., navigating cluttered environments) through fast, robust optimization.

4. Contribution to Signal Processing

A core task in signal processing is designing dynamic filters that adapt to time-varying input signals (e.g., audio, communication signals). MINN's fast convergence and disturbance resilience can address traditional filters' latency and instability issues [R2].

5. Contribution to Control Systems

Real-time state estimation (e.g., for UAV flight control [R3]) is vital for control systems, where latency and robustness impact safety. MINN will outperform traditional methods by maintaining accurate state predictions under noise, which can ensure responsive control for dynamic systems.

6. Contribution to Financial Numerical Analysis

Financial portfolio optimization requires dynamic adjustment to market fluctuations to balance returns and risk [R4]. MINN can leverage its rapid solution capabilities to enable real-time portfolio updates and demonstrates robustness against market disturbances.

The versatility of MINN extends far beyond the current work, with clear potential to impact additional fields:

- **Autonomous driving:** Real-time trajectory optimization under dynamic traffic conditions, where MINN's speed and robustness ensure passenger safety [R5].
- **Industrial process control:** Dynamic adjustment of production parameters to minimize energy consumption and waste, leveraging MINN's ability to handle time-varying constraints.
- **Medical imaging:** Reconstruction of positron emission tomography images by solving time-varying matrix equations. MINN's high accuracy enhances temporal resolution, enabling more precise diagnosis of diseases [R6].

According to the above analysis, the contributions of this paper are not confined to a narrow subfield. Instead, it can be applied to wide fields. As a universal framework for dynamic optimization, MINN aligns with the mission of Nature Communications to publish advances of broad significance to specialists across multiple disciplines.

[R1] Marcucci, Tobia, et al. "Motion planning around obstacles with convex optimization." *Science Robotics* 8.84 (2023): eadf7843.

[R2] Dylov, Dmitry V., and Jason W. Fleischer. "Nonlinear self-filtering of noisy images via dynamical stochastic resonance." *Nature Photonics* 4.5 (2010): 323-328.

[R3] Kim, Taewi, et al. "Wing-strain-based flight control of flapping-wing drones through reinforcement learning." *Nature Machine Intelligence* 6.9 (2024): 992-1005.

[R4] Gunjan, Abhishek, and Siddhartha Bhattacharyya. "A brief review of portfolio optimization techniques." *Artificial Intelligence Review* 56.5 (2023): 3847-3886.

[R5] Geisslinger, Maximilian, Franziska Poszler, and Markus Lienkamp. "An ethical trajectory planning algorithm for autonomous vehicles." *Nature Machine Intelligence* 5.2 (2023): 137-144.

[R6] Ma, Jun, et al. "Segment anything in medical images." *Nature Communications* 15.1 (2024): 654.

Comment 2: *The English is quite poor and there are many typos (e.g., upper case and lower case letters are used here and there). Some sentences are also awkward. The authors are advised to perform a major polishing effort.*

Response. The authors have revised the manuscript again and again to improve the quality of the manuscript. In addition, the authors have asked more than three high level researchers to further polish the manuscript.

Comment 3: *There is no “neural network”, in the sense that there is no training of the network. The “neural network” is designed such that the Lyapunov function decreases. As done in control theory. The neural network is just another non-linear control law.*

Response. The meta-interactive neural network (MINN) proposed in this paper is not the same with the traditional neural network. In fact, the proposed MINN belongs to a recurrent neural network (like Hopfield neural network originally proposed by John J. Hopfield in 1982 who awarded the Nobel Prize in 2024 [R7-R9]).

Recurrent neural networks are artificial neural networks where the computation graph contains directed cycles. Unlike feedforward neural networks, where information flows strictly in one direction from layer to layer, in recurrent neural networks (RNNs), information travels in loops from layer to layer so that the state of the model is influenced by its previous states. While feedforward neural networks can be thought of as stateless, RNNs have a memory which allows the model to store information about its past computations. This allows recurrent neural networks to exhibit dynamic temporal behavior and model sequences of input-output pairs. Due to this characteristic, the RNNs excels at handling time-related tasks. RNNs have also been used in reinforcement learning to solve very difficult problems at a level better than humans. A recent example is AlphaGo, which beat world champion Go player Lee Sedol in 2016.

As a typical recurrent neural network, the Hopfield neural network (HNN) is taken as an example here. In ref. [R8], John J. Hopfield designed an energy function of the system, and he pointed out that the energy function must converge to a stable local optimum value through iterative updates of the neuron states. Notably, HNN is a single-layer network where inter-neuronal connections (analogous to synapses in biological neurons) can be either unidirectional or bidirectional. It is important to clarify that, unlike traditional feed-forward neural networks, HNN does not involve autonomously adjusting its weight matrix through training. Once task requirements, the corresponding number of neurons, and internal attractor criteria (i.e., specific states of each neuron) are defined, the HNN weight matrix can be manually computed via mathematical procedures. After inputting this precomputed weight matrix, the HNN autonomously performs iterative updates to reach predefined attractor states, with the weight matrix remaining unchanged throughout the iteration process.

The MINN proposed in this paper is a brand-new recurrent neural network, which is inspired by the coupled information exchange between different neurons in biological nervous systems, enabling neural networks to possess enhanced information processing capabilities. To further illustrate the architecture of MINN, Fig. 3e in the revised manuscript presents its topology. The activation functions of the proposed MINN used in this paper are Linear activation function, Power activation function, Swish activation function, and ReLU activation function. The learning algorithm of the proposed MINN is a training-free learning approach and is suitable for real-time computation and time-varying solving problems. The proposed MINN neural network learning algorithm works as follows. First, define an unbounded vector-type error function. Second, substitute the error function into the neural dynamics design formula according to the neural dynamics design methodology. Third, compute the real-time state values based on the evolutionary behavior of the dynamic system itself.

In summary, we contend that the proposed MINN in this manuscript aligns with the characteristics of neural networks. The convergence formulas and the overall architecture of MINN are designed by drawing on the Lyapunov stability analysis theorem from control theory, which represents a legitimate integration of neural network design with control-theoretic principles. We have explained the connection between MINN and HNN in the revised version and clarified why parameters do not need to be trained, that is:

“Unlike the network models commonly used in deep learning [34] and machine learning [35], the parameters $\tau(t)$

and matrix α in the proposed MINN (8) do not require training. In other words, MINN employs a training-free learning approach. Once the task requirements and the internal attractor criteria (i.e., the specific state of each neuron) are determined, the weight matrix of the network can be calculated through mathematical procedures. The goal of MINN is to make the neuronal states converge to the expected values through connection weights and mathematical rule. The mathematical basis for this convergence is precisely the Lyapunov stability analysis theorem. The design of connection weights is also determined according to the mathematical rules derived in the next section.”

Fig 3. Topology diagram of the proposed MINN.

[R7] Hopfield, John J. “Pattern recognition computation using action potential timing for stimulus representation.” Nature 376.6535 (1995): 33-36.

[R8] Hopfield, John J. “Neural networks and physical systems with emergent collective computational abilities.” Proceedings of the National Academy of Sciences 79.8 (1982): 2554-2558.

[R9] Hopfield, John J. “Artificial neural networks.” IEEE Circuits and Devices Magazine 4.5 (1988): 3-10.

Comment 4: The “interaction” is rather natural; there is no reason to assume that a diagonal control law would work.

Response. Thank you for your insightful comment regarding the “interaction” in the MINN and the consideration

of diagonal elements. We appreciate your perspective and would like to explain why we believe the discussion on diagonal elements is necessary, from both mathematical and comprehensive parameter analysis perspectives.

1. From a mathematical viewpoint. The design formula of MINN is based on

$$\dot{\varepsilon}_i(t) = -\tau(t)\varepsilon_i(t) + \sum_{j=1}^{\nu} \alpha_{ij} f_j(\varepsilon_j(t)) \quad (1)$$

it is evident that MINN consists of two components: the neuron's own dynamics $\dot{\varepsilon}_i(t) = -\tau(t)\varepsilon_i(t)$ and the interactive coupling term $\sum_{j=1}^{\nu} \alpha_{ij} f_j(\varepsilon_j(t))$. If the activation function is linear, the equation simplifies to

$$\dot{\varepsilon}_i(t) = -\tau(t)\varepsilon_i(t) + \sum_{j=1}^{\nu} \alpha_{ij}(\varepsilon_j(t)). \quad (2)$$

In this scenario, $\tau(t)$ clearly plays a role equivalent to the diagonal element α_{ii} , as demonstrated by the comparative data in Table 2 of the revised manuscript. Specifically, under identical experimental conditions, the convergence time when $\tau(t) = 0$ and $\alpha_{ii} = -4$ is consistent with that when $\tau(t) = -4$ and $\alpha_{ii} = 0$. This result supports our conclusion regarding the connection between the design of diagonal elements in α and the neuron's own dynamic parameter $\tau(t)$, validating the significance of discussing the role of diagonal elements. In practical applications, we can flexibly choose whether to set the diagonal elements to 0 as needed, a point explicitly clarified in the revised manuscript.

2. From the perspective of comprehensive parameter analysis. The key parameters of MINN include $\tau(t)$ and α . To ensure an unbiased and thorough discussion of their roles, we compared MINN with other models under the following scenarios:

- 1) $\tau(t) = 0$ with fixed α , to analyze the role of the neuron's own dynamic parameter $\tau(t)$.
- 2) $\tau(t) = 0$ with varying diagonal elements α_{ii} , to examine the role of α_{ii} .
- 3) $\tau(t) = 0$ with $\alpha_{ii} \neq 0$ and varying non-diagonal elements α_{ij} , to explore the role of α_{ij} .
- 4) $\tau(t) \neq 0$ with $\alpha_{ii} = 0$ and varying non-diagonal elements α_{ij} , to explore the role of α_{ij} .

Notably, cases 3) and 4) are compared to reveal the connection between diagonal elements α_{ii} and $\tau(t)$. In the revised manuscript, we have corrected a typo in Figs. 6c and 6d by changing the legend label from α_{ii} to α_{ij} . Additionally, Table 2 has been redesigned to better illustrate the impact of parameter configurations on convergence time, with the revised text stating:

“The convergence times of the system under Cases 2 and 3 are recorded in Table 2. Additionally, an extra case is recorded, i.e., $\tau(t) = -4$, $\alpha_{ii} = 0$. In this case, the convergence time of the system is consistent with Case 3, as when $\tau(t) = -4$, $\tau(t)$ can serve as a diagonal element α_{ii} . Therefore, it is evident that when $\tau(t) = e^t$ and $\alpha_{ii} = 0$, the system will achieve better convergence performance. When using it in practice, it is possible to consider whether to retain the value of the diagonal element based on specific needs. In summary, this example presents various parameter combinations and provides a detailed explanation of the underlying logic by which parameters influence convergence time. This will serve as a valuable resource for further research by subsequent scholars.”

Comment 5: *When one observe the contribution as it really is, then there is much literature that is missing and methods to compare to.*

The work of Wei Ren in this domain is quite extensive (both quadratic and non, both distributed and centralized, both unconstrained and constrained). For example, “Distributed Continuous-Time Algorithms for Optimal Resource Allocation With Time-Varying Quadratic Cost Functions”, IEEE Transactions on Control of Network Systems,

TABLE I: Time taken for $\|\varepsilon(t)\|_2 = 0.01$ and $\|\varepsilon(t)\|_2 = 0.001$ under linear activation function for five cases

		α_{ii}					
		-4	-5	-6	-7	-8	-10
$\tau(t) = 0$	$\ \varepsilon(t)\ _2 = 0.01$	3.69s	2.31s	1.7s	1.35s	1.13s	0.86s
	$\ \varepsilon(t)\ _2 = 0.001$	can't	3.17s	2.31s	1.83s	1.51s	1.14s
		$\alpha_{ij}, i \neq j$ (Initial values with same signs)					
		0.1	0.2	0.3	0.4		
$\tau(t) = 0, \alpha_{ii} = -4$	$\ \varepsilon(t)\ _2 = 0.01$	2s	2.32s	2.91s	3.99s		
	$\ \varepsilon(t)\ _2 = 0.001$	2.64s	3.11s	3.95s	can't		
		$\alpha_{ij}, i \neq j$ (Initial values with opposite signs)					
		0.1	0.2	0.3	0.4		
$\tau(t) = 0, \alpha_{ii} = -4$	$\ \varepsilon(t)\ _2 = 0.01$	1.85s	1.83s	1.8s	1.75s		
	$\ \varepsilon(t)\ _2 = 0.001$	2.5s	2.45s	2.41s	2.38s		
		$\alpha_{ij}, i \neq j$ (Initial values with same signs)					
		0.1	0.2	0.3	0.4		
$\tau(t) = -4, \alpha_{ii} = 0$	$\ \varepsilon(t)\ _2 = 0.01$	2s	2.32s	2.91s	3.99s		
	$\ \varepsilon(t)\ _2 = 0.001$	2.64s	3.11s	3.95s	can't		
		$\alpha_{ij}, i \neq j$ (Initial values with opposite signs)					
		0.1	0.2	0.3	0.4		
$\tau(t) = -4, \alpha_{ii} = 0$	$\ \varepsilon(t)\ _2 = 0.01$	1.85s	1.83s	1.8s	1.75s		
	$\ \varepsilon(t)\ _2 = 0.001$	2.5s	2.45s	2.41s	2.38s		

2021. “Distributed continuous-time time-varying optimization for networked Lagrangian systems with quadratic cost functions”, *Automatica*, 2025. But there are many others.

Response. Thanks to the reviewers for providing these excellent references. We have discussed and cited these outstanding papers in the Introduction section as refs. [3, 6, 7]. In addition, we have added a comparison of methods in the Introduction section, that is

“Comparison with related works: The MINN proposed in this paper differs from most existing models. Specifically, traditional research focuses on improving network performance by adjusting parameters and activation function [12, 18-21]. In contrast, MINN enhances computational efficiency by constructing communication topologies between neurons and leveraging the collective dynamics of neurons. This novel approach provides a new perspective for existing research; based on MINN, better models are expected to be derived by adjusting network structures and coupling parameters. Furthermore, we note that the design of activation functions in refs. [11, 12, 16-18] is relatively simplistic, taking monotonically increasing odd functions as the sole criterion for selecting activation functions. Although this assumption simplifies mathematical proofs, its application scope is narrow. Through mathematical derivation and proof, this study expands the application scope of activation functions. In refs. [28-31], neural network models similar to MINN are used to solve control problems of nonlinear systems. However, it is evident that these neural networks are only applied to specific problems. The MINN proposed in this paper is a more general method. In addition to solving TVQP problems, it can be applied to other time-varying problems (e.g., the Sylvester equation and robot motion planning, as detailed in the Supplementary Information).”

Comment 6: *Some interesting avenues for research are left unexplored. For instance, in the spirit of a neural network, it would be interesting to learn the coupling matrix α , instead of imposing it.*

Response. The authors thank the reviewer for his/her valuable suggestion regarding potential research. We fully acknowledge the significance of this direction and appreciate the insight in highlighting it.

This study focuses on establishing the fundamental framework of MINN and verifying its effectiveness in solving time-varying problems. The design of the coupling matrix α through theoretical derivation (ensuring diagonal dominance and specific sign conditions) served to provide a rigorous mathematical foundation for guaranteeing convergence, which was critical for demonstrating the core mechanism of MINN. This approach allowed us to clearly verify the contribution of the network's topological structure to computational efficiency.

We entirely agree that exploring the learning of α represents a highly promising extension of this research. Such a direction would further bridge the gap between our model and adaptive neural network paradigms, potentially enhancing flexibility in handling complex, unstructured problems. In future work, we plan to investigate incorporating learning mechanisms for α , possibly leveraging reinforcement learning or adaptive control frameworks to optimize the coupling matrix dynamically based on problem characteristics. This could involve designing loss functions that balance convergence speed, stability, and computational efficiency, while maintaining the theoretical guarantees established in current analysis.

Finally, the authors would like to say thanks again sincerely to the reviewer for his/her time and efforts spent in reviewing the manuscript, as well as many constructive comments he/she has provided which have improved much further the presentation and quality of this manuscript.

Summary of Changes

Revision Information (#Ref. No. NCOMMS-25-27314B)

“A Meta-Interactive Neural Network for Solving Time-Varying Quadratic Programming Problems”

Zhijun Zhang, Xiangliang Sun, Yiqi Liu and Yamei Luo

Responses to Reviewer 3

Comment: *Dear authors. Thanks for the revised version and the attention you have put in replying to my comments. I appreciate the effort and the contribution of your paper, but I do not think it is Nature material. I think it should be going through the scrutiny of a specialised journal, where the maths and the contribution to the neural network community can be thoroughly checked. Time-varying quadratic programming, as you have mentioned, is a topic of growing interest; many methods have been proposed – yours, while interesting, is not a disruptive advance that would make a difference in the broad community.*

Response. We are deeply grateful for the time and expertise you have invested in evaluating our work. We highly value your suggestions, in our future endeavors, we plan to explore extended research on MINN and, as you recommended, submit to specialized journals in the fields of neural networks or optimization.

Responses to Reviewer 4

Comment: *All the comments have been addressed. The idea is very **novel**, the theory of the paper is **rigorous** and the experiments are **solid**. It makes **significant** contributions to a wide range of fields. I suggest it be **accepted** for publication directly.*

Response. The authors sincerely thank the reviewer for his/her recognition and praise.

Responses to Reviewer 5

Comment: The paper has been carefully revised in response to the reviewers’ concerns and is now ready for acceptance.

Response. The authors sincerely thank the reviewer for his/her recognition and acceptance of the manuscript.

Finally, the authors would like to say thanks again sincerely to the reviewer for his/her time and efforts spent in reviewing the manuscript, as well as many constructive comments he/she has provided which have improved much further the presentation and quality of this manuscript.